# Impact of keratocyte differentiation on corneal opacity resolution and visual function recovery in male rats

Andri K. Riau [1,2,6], Zhuojian Look [1,2,6], Gary H. F. Yam[1,3], Craig Boote[4], Qian Ma [4], Evelina J. Y. Han[1], Nur Zahirah binte M. Yusoff[1], Hon Shing Ong [1,2,5], Tze-Wei Goh[1], Nuur Shahinda Humaira binte Halim[1] & Jodhbir S. Mehta [1,2,5] ✉

Intrastromal cell therapy utilizing quiescent corneal stromal keratocytes (qCSKs) from human donor corneas emerges as a promising treatment for corneal opacities, aiming to overcome limitations of traditional surgeries by reducing procedural complexity and donor dependency. This investigation demonstrates the therapeutic efficacy of qCSKs in a male rat model of corneal stromal opacity, underscoring the significance of cell-delivery quality and keratocyte differentiation in mediating corneal opacity resolution and visual function recovery. Quiescent CSKs-treated rats display improvements in escape latency and efficiency compared to wounded, non-treated rats in a Morris water maze, demonstrating improved visual acuity, while stromal fibroblasts-treated rats do not. Advanced imaging, including multiphoton microscopy, small-angle X-ray scattering, and transmission electron microscopy, revealed that qCSK therapy replicates the native cornea's collagen fibril morphometry, matrix order, and ultrastructural architecture. These findings, supported by the expression of keratan sulfate proteoglycans, validate qCSKs as a potential therapeutic solution for corneal opacities.

Corneal haze or opacity refers to a loss of corneal transparency which is essential for visual function. Opacification of the cornea limits its ability to accurately transmit and focus light[1]. Corneal opacities can arise from a wide range of pathologies and aetiologies including trauma, infections, and autoimmunity, and often present as corneal scarring (the 4th leading cause of blindness in the world)[2]. This opacity can be partially attributed to the disruption in the normal function of native corneal stromal keratocytes (CSKs)—the main resident cells of the corneal stroma—as they normally express crystallin proteins that confer transparency, such as aldehyde dehydrogenases, ALDH1A1 and ALDH3A1[3,4]. Furthermore, CSKs are responsible for the tight regulation of the cornea stromal milieu, secreting extracellular matrix (ECM)

components, such as collagen types I and V, as well as keratan sulfate (KS) proteoglycans, such as keratocan and lumican[3,5,6]. These active processes in the stromal microenvironment allow for the formation of interweaving orthogonally organized corneal lamellae containing collagen fibers of uniform size and spacing[7], thereby giving rise to the cornea's optical transmittance and refractive properties.

In the pathogenesis of corneal opacification, injury disrupts the basement membrane, allowing profibrotic cytokines and chemokines, for example, interleukin-1α (IL-1α), transforming growth factor-β1 (TGF-β1) and platelet-derived growth factor (PDGF), to diffuse into the stroma[8,9]. CSKs exposed to trauma undergo apoptosis or necrosis, while surviving CSKs that are exposed to TGF-β1 and other profibrotic

[1]Tissue Engineering and Cell Therapy Group, Singapore Eye Research Institute, Singapore 169856, Singapore. [2]Ophthalmology and Visual Sciences Academic Clinical Programme, Duke-NUS Medical School, Singapore 169857, Singapore. [3]Department of Ophthalmology, University of Pittsburgh, Pittsburgh, PA 15213, USA. [4]School of Optometry and Vision Sciences, Cardiff University, Cardiff CF24 4HQ, UK. [5]Corneal and External Eye Disease Department, Singapore National Eye Centre, Singapore 168751, Singapore. [6]These authors contributed equally: Andri K. Riau, Zhuojian Look. ✉e-mail: jodhbir.s.mehta@singhealth.com.sg

signaling molecules lose their functional phenotypes and differentiate into corneal stromal fibroblasts (SFs) and myofibroblasts (myoSFs)[9–11]. The SFs and myoSFs secrete abnormal ECM components that are critical for stromal wound healing[6]. The SFs are also responsible for recruiting inflammatory cells to the injury site by sending pro-inflammatory signals[11]. Together with edema, infiltration of inflammatory cells, a loss of corneal stromal matrix homeostasis, and the decreased production of crystallins and normal stromal matrix components by myoSFs, the microstructure of the cornea is severely dysregulated, scattering instead of refracting incident light[12].

Clinically, options for the treatment of corneal opacities are limited but can range from conservative to invasive therapies. Conservatively, watchful waiting can be recommended particularly in opacities secondary to corneal scars that are asymptomatic and in the peripheral cornea. Other therapies employed are vision correction through scleral lenses, prescription glasses or rigid gas permeable (RGP) contact lenses, and photo-therapeutic keratectomy—a form of laser surgery[13,14]. Corneal transplantation, such as a deep anterior lamellar keratoplasty (DALK) or penetrating keratoplasty (PK) can be performed for severe opacities[15]. However, the long-term success of surgical transplantation of allogeneic corneal tissue is associated with major limitations, chiefly that of low donor supply and the requirement for strict adherence to topical immunosuppression to prevent graft rejection, with some studies reporting a rejection frequency of up to 41% in PK[16–18].

Recently, cell-based therapies for corneal stromal regeneration have emerged as a promising therapeutic alternative for the resolution of corneal opacification[9]. Cell-based therapies, in particular, cell injections into the stroma of the cornea have demonstrated varied success in the literature[19,20]. In our previous work, we developed a protocol for the propagation of quiescent CSKs (qCSKs) in an "activated" CSK (aCSK) state from donor corneas[21]. We then demonstrated in a pilot study, that the intrastromal injection of qCSKs into naïve rat corneas had a low incidence of adverse reactions or significant side effects. Importantly, qCSK injection did not induce prolonged immune responses[22]. The ability to cultivate qCSKs, while challenging, greatly reduces the supply constraints that current surgical treatments utilizing transplantation face. Also, the minimally invasive nature of intrastromal cell injection with a pinpoint entry site is easier to perform and potentially presents with fewer postoperative complications than keratoplasties, such as DALK or PK.

However, to date, it is unknown if utilizing qCSKs, aCSKs, or SFs for intrastromal cell injection therapy in a rat acute haze model contributes to significantly differential outcomes and mechanisms on corneal haze resolution and visual function recovery. This is a key next step following our earlier study in quantitatively characterizing the cellular phenotype and gene expression profiles of qCSKs, aCSKs, and SFs. Of particular interest, aCSKs are readily propagated in cell culture and do not require a 14-day stabilization period in serum-free media. Proving the viability of utilizing aCSKs for cell injection would potentially reduce the culture expansion complexity and increase the treatment pool potential from a single donor cornea harvest[21]. Additionally, there is also limited work in categorically defining the quality of cell injection delivery techniques, as well as how said quality affects treatment efficacy independent of the cell types chosen.

In this study, we used human same-donor-derived qCSKs, aCSKs, and SFs, with an intrastromal injection delivery methodology, in male rats (*Rattus norvegicus*) with acute and chronic corneal haze that was induced by irregular phototherapeutic keratectomy (IrrPTK) with an excimer laser. To determine if the choice of cell type–which is derived by supplementation of different concentrations of serum during cell propagation–has significant differential effects on treatment efficacy, we looked at the degree of opacification resolution, collagen fibril characteristics, ECM protein expression, as well as visual function recovery. We also determined a metric for classifying the efficacy of

cell injection into categorically good or poor procedures, with several observable visual signs. Our results demonstrated that cell delivery quality and cell type choice had significant effects on corneal haze resolution and visual function recovery, with qCSKs coupled with good quality injections providing the best results. The effect of delivery quality on haze reduction was significant after post-procedure day 21, suggesting that density and spatial localization of the injected cells was of critical importance for long-term outcomes. We further demonstrated that the qCSKs resolved the haze by restoring the collagen fibrillar organization and KS proteoglycans that were disrupted by the laser injury. Finally, we demonstrated that the qCSK therapy was also efficacious in resolving chronic stromal opacity that featured severe vascularization in the cornea, although the potency was marginally attenuated from that in the acute condition.

## Results

### Cultivated keratocyte phenotypes closely resemble native corneal stromal keratocytes

Morphologically, the cultured qCSKs, following a 14-day stabilization in the serum-free full media, resembled the closest to the 'bona fide' native CSKs (Fig. 1a). The qCSKs presented stellate morphology with thin and rounded cell bodies and extended cell processes. This was in contrast with proliferative SFs, which had a bipolar morphology with large and elongated cell bodies and pseudopodial processes. The SFs also preferred to grow adjoining to one another compared to the qCSKs, which preferred to grow in a spatially sparse fashion. Most aCSKs had a similar morphology to the qCSKs but some cells assumed a shorter and more proliferative morphology. The qCSKs expressed higher levels of typical CSK-associated genes in the corneal stromal tissue than aCSKs or SFs (Fig. 1b and Supplementary Table 1)[3,21,23]. The expression of *ALDH1A1* ($p = 0.016$), *ALDH3A1* ($p = 0.006$), lumican (*LUM*) ($p = 0.036$), keratocan (*KERA*) ($p = 0.008$), β1-3-N-acetylglucosaminyltransferase 7 (*B3GNT7*) ($p = 0.007$), and collagen 8A2 (*COL8A2*) ($p = 0.040$) genes in the qCSKs was significantly higher than the SFs. The transcription level of the carbohydrate sulfotransferase 6 (*CHST6*) gene was also higher in the qCSKs than the SFs, although the difference did not reach a significant level ($p = 0.114$). The aCSKs exhibited an intermediate level of the above-mentioned genes, suggesting a correspondingly intermediate state between qCSKs and SFs.

The unique markers of native CSKs are the dual expression of KS proteoglycans, keratocan and lumican[5]. This dual expression differentiates CSKs from progenitor corneal stromal stem cells (CSSCs), which do not express keratocan[24]. We demonstrated an abundant expression of keratocan (Fig. 1c) and lumican (Fig. 1d) in the cytoplasm of the cultured qCSKs, complementing the gene expression analysis. Phalloidin staining demonstrated a cortically diffuse and unpolymerized pattern of F-actin distribution in the qCSKs. In contrast, F-actin in SFs showed parallel intracellular alignment and a higher degree of polymerization (Fig. 1c). The staining substantiated the bright field microscopy observation earlier (Fig. 1a), revealing the transformation of some of the aCSKs into an SF-like appearance with the presence of intracellular stress fibers. These aCSKs expressed relatively weaker keratocan and lumican than the qCSKs (Fig. 1c, d). The SFs, all of which featured stress fibers extending to the pseudopodia, negligibly expressed keratocan and lumican (Fig. 1c, d).

Given that the propagation and cultivation of aCSKs in 0.5% serum is both time-consuming and laborious for cell manufacturing, we aimed to expedite the process by culturing donor-derived stromal cells in 10% serum, followed by serum starvation for 14 days in serum-free complete media, in an effort to generate qCSKs. Our findings revealed that these serum-starved SFs had similar levels of *ALDH1A1* ($p = 0.999$), *CHST6* ($p = 0.150$), and *LUM* ($p = 0.147$) to the SFs (Supplementary Fig. 1a). Notably, *ALDH3A1* and *KERA* expression in the serum-starved SFs was markedly higher than that in SFs

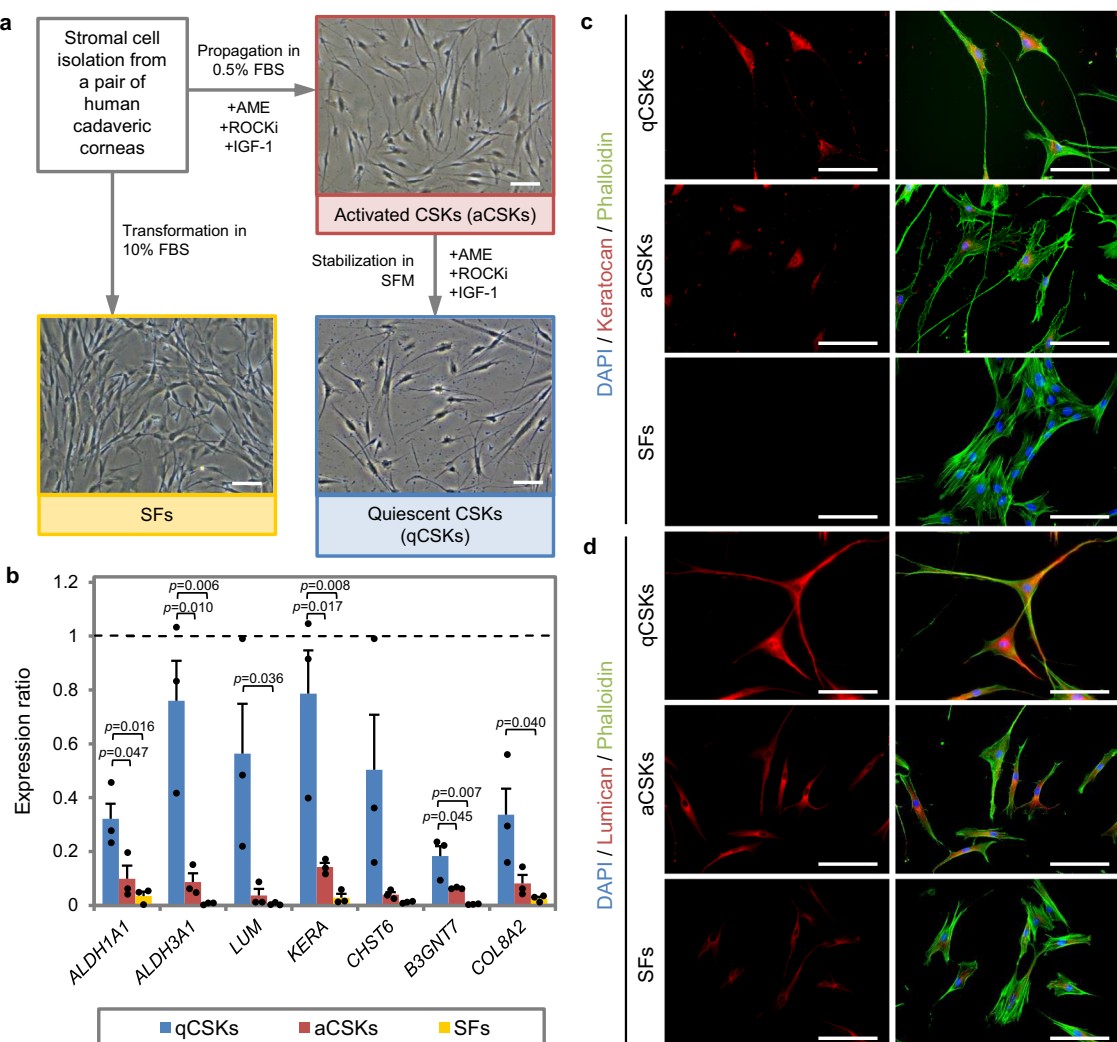

**Fig. 1 | Phenotypical features of cultivated corneal stromal keratocytes. a** The propagation of corneal stromal keratocytes (CSKs) was achieved by culturing the corneal stromal cells in the "activated" form in the 0.5% fetal bovine serum (FBS)-supplemented media, containing amniotic membrane extract (AME), ROCK inhibitor (ROCKi), and insulin growth factor 1 (IGF-1). The "activated" state was referred to as the activated CSKs (aCSKs). The aCSKs entered a quiescent state following culture in serum-free media (SFM), which we referred to as the quiescent CSKs (qCSKs). The qCSKs featured stellate morphology with thin cell bodies and long cell processes. As a comparison, stromal fibroblasts (SFs) that were transformed by culturing the cells in 10% serum-supplemented media had a bipolar morphology with large cell bodies and pseudopodial processes. **b** The qCSKs (blue bars)

expressed higher levels of native CSK's genetic markers than the aCSKs (red bars) and SFs (yellow bars). The gene expression levels were normalized to the corneal stromal tissue, indicated by the dashed line. Data were presented as mean ± SEM ($n = 3$ in each group). Statistical significance was assessed with one-way ANOVA, followed by post hoc Tukey test. Immunocytochemistry of keratocan (**c**) and lumican (**d**), double stained with phalloidin, confirmed the earlier morphological and gene expression analysis. The qCSKs exhibited more abundant keratocan and lumican proteins than aCSKs. In contrast, the SFs were largely absent of these proteoglycans. The staining was repeated on 3 independent cells. Scale bars = 50 μm. Source data are provided as a Source Data file.

(both $p = 0.004$) and *B3GNT7* was undetectable in both cell types. In terms of morphology, the serum-starved SFs appeared to be in an intermediary state between SFs and qCSKs, characterized by long cell processes but with larger cell bodies, and bipolar and parallel orientation (Supplementary Fig. 1b). Consistent with the gene expression, the protein expression of lumican and keratocan was absent but the ALDH3A1 exhibited increased expression in the serum-starved cells as compared to SFs (Supplementary Fig. 1c). Due to the inability to recapitulate qCSK morphology and phenotypes, serum-starved SFs were deemed unsuitable as an alternative to qCSKs.

### Cell delivery quality dictates the therapeutic efficiency of cultivated keratocytes

Following our pilot study[22], we attempted to validate the outcomes of the qCSK therapy in rats with IrrPTK-induced acute corneal haze.

The cell injection was administered 7 days after the laser injury (also referred to as post-injection day 0 or PID0) and the outcome was monitored for 21 days (PID21) (Supplementary Fig. 2a). However, we found variability in the haze clearance efficiencies in these rats. We could be certain that it was not due to the cell death induced by shearing force during the dispensing of the cells through a 31G needle. We have previously shown that >96% of the qCSKs were viable when they were ejected through 27, 30, and 31G needles[22]. Analyzing 18 surgery videos (representative still images of the surgery could be found in Supplementary Fig. 2a and videos in Supplementary Movie 1–4) and then correlating them with the treatment outcomes, we found that the therapeutic efficiency was highly dependent on the efficacy of the qCSK intrastromal injection. We categorized the delivery quality as good if it satisfied all the following criteria (represented by 9 rats): Single injection entry point; and formation of cells-containing bleb at the center of the cornea,

occupying >17% of the corneal area (typically resulting in 3.8x–4.8x volume expansion) (Supplementary Fig. 2a and Supplementary Movie 1). Whereas the delivery quality was considered poor when it fulfilled one or more of the following criteria (represented by the remaining 9 rats): Multiple needle entry points (tunnel collapsed due to softer epithelium, requiring the creation of another injection point) (Supplementary Movie 2); needle-perforated cornea; formation of decentered bleb (Supplementary Movie 3); and backflow of injected cells (Supplementary Movie 4), indicated by a smaller bleb that occupied ≤17% of the corneal area and 1.2x–3.3x volume expansion (Supplementary Fig. 2a). We observed that corneas that received poorly delivered qCSKs had haze and neovascularization (NV) that persisted at almost the same level over 21 days after cell injection. In contrast, the haze, NV, and corneal edema steadily reduced over time in the corneas that received good quality qCSK delivery (Supplementary Fig. 2b–h and Supplementary Tables 2–5). A detailed description of the slit-lamp photography, in vivo confocal microscopy (IVCM), and changes in central corneal thickness (ΔCCT) can be found in the Supplementary Discussion.

### The differentiation state determines the efficacy of cultivated keratocytes in corneal stromal cell therapy
The preceding experiment showed us that the impact of injection quality due to procedural variation could introduce outcome variability. We then sought to understand whether the choice of cell type (qCSKs, aCSKs, or SFs) could affect the therapeutic outcomes of cell therapy. We first compared the qCSK injection outcomes to aCSK and SF injections in naïve corneas to establish the respective cells' baseline safety profiles and clinical effects (Supplementary Fig. 3a). In a naïve state, the qCSKs and aCSKs appeared to have similar safety profiles and biological effects (Supplementary Fig. 3b–h and Supplementary Tables 6–9), which gave us an impetus to study whether aCSK therapy would be feasible in corneas with stromal injury. The other impetus was that the aCSK cultivation was less challenging and time-consuming (because of the exclusion of the stabilization step) than the qCSKs. A detailed description of the outcomes of the investigation can be found in the Supplementary Discussion.

We, again, employed the IrrPTK-induced acute corneal haze model in the next set of experiments and compared the therapeutic efficacy of qCSKs to aCSKs, SFs, and non-treated corneas (Fig. 2a). As expected, the non-treated injured corneas exhibited persistent severe haze and NV at all follow-up time points (Supplementary Fig. 4). Blood vessels could be consistently seen in the IVCM images 14 days after injury. The qCSK therapy outcomes were in line with the haze recovery efficiency in rats that received good qCSK intrastromal delivery (Fig. 2b), where we showed, again, the steady reduction of haze over 21 days and reduction of NV at PID21 following the peak of severity at PID14 (Fig. 2c, d and Supplementary Table 10). Although the aCSK injection did not cause notably worse clinical scores and largely followed the recovery trend of qCSK therapy, it displayed higher grades of haze and NV and total score at all follow-up time points (Fig. 2c–e). For example, on day 21, the median haze, NV, and total scores of the qCSKs group were 0.5 (interquartile range or IQR = 0), 0.5 (IQR = 1.75), and 1 (IQR = 1.75), respectively. The aCSK injection resulted in a median haze, NV, and total scores of 1 (IQR = 0.325), 1.5 (IQR = 1), and 2.25 (IQR = 0.875). The SF injection outcomes, on the other hand, were closer to the state of the non-treated corneas. Although there was a marginal haze reduction at PID14 and 21, the severity was still ≥1 grade higher than the qCSKs group (Fig. 2b, c). The less effective recovery was due to the increasing NV severity (Fig. 2b, d). The total scores of the SFs and non-treated groups were significantly higher than the qCSKs group ($p = 0.031$ and $p = 0.004$, respectively) (Fig. 2e and Supplementary Table 10).

Consistent with the clinical grading, the haze density after qCSK, aCSK, and SF injections reduced over time but the recovery efficiency was higher in the qCSKs group (Fig. 2f and Supplementary Table 11). At PID21, the haze density in the aCSKs group was substantially lower than in the SFs group ($p = 0.031$). The non-treated corneas showed maintenance of high levels of haze density throughout the follow-up periods. The haze density of the non-treated corneas was significantly greater than the aCSKs group at PID14 ($p = 0.032$) and PID21 ($p = 5.42 \times 10^{-5}$). From a haze area viewpoint, the SFs and non-treated groups demonstrated little changes from PID0 to PID21 (Fig. 2g and Supplementary Table 12). The non-treated corneas showed a reduction of the haze area by only $7.24 \pm 9.54\%$, whereas the haze area worsened in the SF-administered corneas by $1.57 \pm 6.48\%$ The haze areas of both groups were remarkably greater than that after the qCSK injection at PID21 ($p = 0.002$ vs. SFs and $p = 0.045$ vs. non-treated). Consistent with the clinical scores, the aCSK injection produced haze area reduction in between the qCSK and SF therapy outcomes. Despite the dissimilar haze recovery profiles, the % ΔCCT differences were almost negligible between qCSKs, aCSKs, and SFs groups (Fig. 2h and Supplementary Table 13). The CCT in the non-treated corneas also appeared to lower after peaking at day 14 post-IrrPTK (PID7 in the treatment arm) but the rate of haze reduction was slower than the corneas treated with any of the corneal stromal cell types.

### Cultivated keratocytes resolve corneal opacity via the restoration of proteoglycans and collagen fibrillar organization
The immunofluorescence of key ECM components in the formation of corneal haze, namely Thy-1[25], α-small muscle actin (α-SMA)[26], fibronectin[27], and collagen 3A1[28], revealed that SFs (Thy-1-positive cells) and myoSFs (α-SMA-positive cells) were largely absent in the naïve and CSK-injected corneas (Fig. 3, top two panels). The non-treated corneas expressed α-SMA, limited to the anterior stroma, but did not express Thy-1 in any layers of the stroma. We detected the presence of Thy-1- and α-SMA-positive cells in the anterior third of the stroma of both the aCSK- and SF-injected corneas. As expected, the naïve corneas did not express fibronectin and collagen 3A1 (Fig. 3, top third and fourth panel). A relatively low expression of fibronectin was deposited in the mid-stroma region of the qCSK-treated corneas. Collagen 3A1 was not observed to be expressed in these corneas at this time point. Interestingly, the aCSK-treated corneas were immunopositive for fibronectin and collagen 3A1, although their presence was not as abundant as in the SF-injected corneas.

KS and KS proteoglycans, namely keratocan and lumican, were abundantly present in the anterior stroma and gradually diminished in the posterior stroma of the naïve corneas (Fig. 3, bottom three panels). The non-treated rats presented a contrasting finding, where the KS and KS proteoglycans were mostly absent in the stroma. Following aCSK and SF injections, the KS and KS proteoglycans were re-expressed in the anterior stroma albeit at lower levels than the naïve corneas. The corneas that received qCSK injections appeared to re-express KS, keratocan, and lumican close to the naïve corneal state.

Because the aCSKs produced therapeutic efficiency that was inferior to the qCSKs and low-level fibrotic responses in the acute opacity cases, we did not think the cells were suitable for corneal stromal cell therapy and therefore, the aCSK group was excluded in the next series of experiments. To further elucidate the haze clearance mechanism of cultivated qCSKs in vivo, we scanned the corneas with small-angle X-ray scattering (SAXS) (Fig. 4a). The scans revealed a ~4.8 nm increase in the mean interfibrillar distance (IFD) ($p = 0.011$) and ~1.1 nm increase in the fibril diameter (FD) ($p = 0.022$) in the qCSK-injected corneas relative to the naïve corneas (Fig. 4b, c and Supplementary Table 14). The non-treated corneas also displayed elevated IFD and FD at the same time point. The SF-injected corneas, on the other hand, had a similar IFD and FD as the naïve corneas. Besides IFD and FD, the SAXS also revealed the degree of collagen matrix order[29]. Boote and colleagues demonstrated with SAXS that corneal injuries

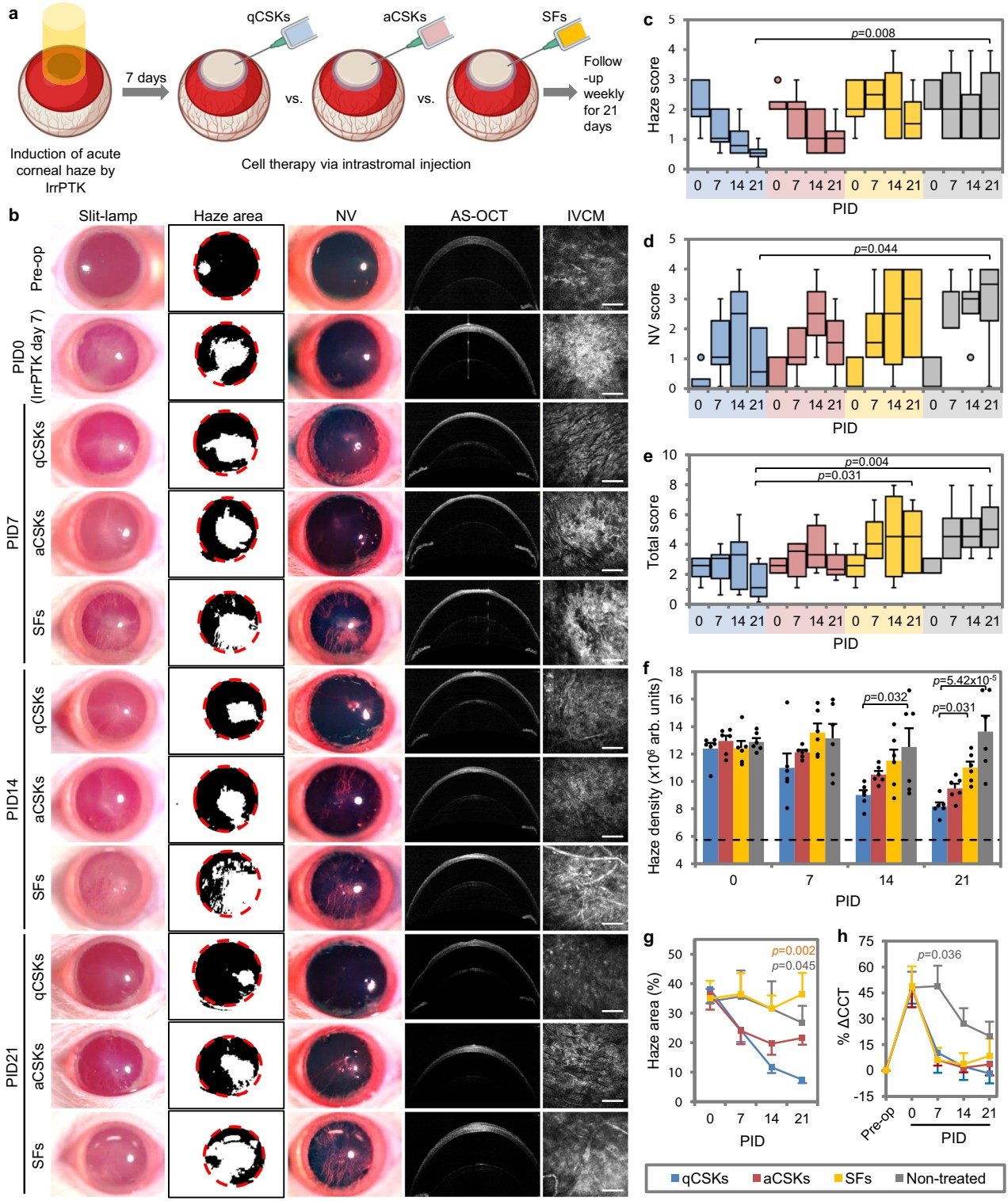

akin to IrrPTK, which disrupt the epithelium and Bowman's membrane, resulted in more intense haze due to a lower matrix order than debrided corneas with intact Bowman's membrane and naïve corneas[29]. We could see that the qCSKs returned the matrix order to the naïve corneal state but not the SFs (Fig. 4d and Supplementary Table 14). We did not find any significant difference in the matrix order between the qCSK-injected and naïve corneas ($p = 0.804$). In contrast, the SF treatment resulted in a similar degree of fibrillar order to the non-treated corneas, which was significantly lower than the naïve corneas ($p = 0.026$).

The transmission electron microscopy (TEM) of the corneas supported the SAXS findings, where we showed that, at lower magnifications, uniformly organized collagen lamellae were observed in the anterior stroma of the naïve corneas and were also similarly observed in the qCSK-treated corneas (Fig. 4e, top panel). In these corneas, the lamellae contained longitudinally arranged collagen fibrils that appear in darker bands, alternating with cross-sectionally arranged collagen fibrils appearing in lighter bands, in a more regular fashion. In the SF-injected and non-treated injured corneas, the alternating dark and light bands were not that obvious in some regions. At higher

**Fig. 2 | Postoperative outcomes of cell injections in corneas with excimer laser-induced acute opacity. a** Experimental setup and timeline. The treatment outcomes were also compared to the non-treated corneas (see Supplementary Fig. 4 for the imaging results). **b** The corneal imaging showed that quiescent corneal stromal keratocytes (qCSKs) had the best therapeutic efficiency, followed by activated CSKs (aCSKs) and then stromal fibroblasts (SFs). SF injection also produced haze reduction albeit in a less efficient manner compared to the qCSKs, which was most likely caused by the increasing neovascularization (NV). The non-treated corneas exhibited persistently severe corneal haze and NV at any time point. The slit-lamp observation was supported by the haze (**c**), NV (**d**), and total grades (**e**). In the box plots, the center line shows the median score; box limits show the 1st and 3rd quartile; whiskers show minimum and maximum values; and points indicate outliers. Statistical significance was assessed with Kruskal–Wallis, followed by post hoc Dune-Bonferroni test. **f** The opacity was substantiated by the IVCM-based haze density measurements. The dash line indicates the mean of haze density pre-operatively. **g** From a haze area point-of-view, the SFs and non-treated groups demonstrated almost no changes from PID0. The qCSKs showed the highest efficiency in haze area reduction, followed by aCSKs. The significant $p$ values were relative to the qCSKs. **h** The % $\Delta$CCT over time did not differ in the qCSKs, aCSKs, and SFs groups. On the other hand, the CCT reduction was also observed in the non-treated corneas, albeit at a lower rate. The significant $p$ value was relative to the qCSKs. Data are presented as mean ± SEM. Statistical significance was analyzed with one-way ANOVA, followed by post hoc Tukey test. The qCSKs, aCSKs, SFs, and non-treated are represented by blue, red, yellow, and gray in the box plots, bar graphs, and line graphs. $n = 6$ rats in each group. Scale bars = 100 μm. Source data are provided as a Source Data file.

magnification, especially in lamellae with longitudinally appearing collagen fibrils, we could see that the fibrils were relatively shorter than those in the naïve and qCSK-injected corneas (Fig. 4e, bottom panel). The lamellae of SF-injected and non-treated groups were also interrupted by the cross-sectionally appearing collagen fibrils.

We then employed multiphoton microscopy to further characterize the collagen fibril morphometry and organization in the rat corneas. The second harmonic generation (SHG) signals revealed a similar tissue area ratio (TAR), collagen area ratio in tissue (CART), and collagen fibrillar density (CFD) between naïve and qCSK-injected corneas (Fig. 4f–i and Supplementary Table 15). TAR, a measure for tissue integrity[30], of the corneas in the naïve group, was superior to the SFs ($p = 0.030$) and non-treated ($p = 0.061$) groups (Fig. 4g). CART and CFD, refer to the relative amount of collagen in the corneal tissue and how closely packed the collagen fibers are in the tissue. CART and CFD following SF injections were comparable to the non-treated corneas and were significantly lower than the naïve tissues ($p = 0.002$ and $p = 0.006$, respectively) (Fig. 4h, i). Consistent with the CART and CFD, the collagen fiber count (CFC) per mm$^2$ of qCSK-injected corneas was comparable to the non-treated corneas (Fig. 4j). On the other hand, the CFC was 39.99 ± 6.50% and 24.96 ± 9.01% lower in the SF-treated and non-treated corneas, respectively.

In line with the TEM observation, we found that the collagen fiber length (CFL) in the qCSK-injected corneas was comparable to the naïve tissues (Fig. 4k). The corneal CFL in the SFs and non-treated groups were 7.32 ± 3.30% and 9.11 ± 2.56% shorter, respectively than in the naïve group. The collagen area reticulation density (CARD), an indication of the degree of fiber branching, a feature of the structural integrity in the healthy cornea[7], following qCSK injections, was comparable to the naïve corneal state (Fig. 4l). The CARD in the SF-treated and non-treated corneas tended to be lower than in the naïve and qCSKs groups but the difference was not statistically significant. Aggregating all the above parameters in a radar map and normalizing them to the naïve cornea values, we could see the collagen fiber profile in the qCSK-treated corneas recapitulated the best of the naïve corneas (Fig. 4m). By measuring the pseudohexagonal area, the qCSK-administered corneas assumed approximately 103% of the naïve corneal collagen profile. The SF-treated corneas displayed a collagen fiber profile that was closer to the non-treated corneas and was only ~33% of the naïve state. The collagen fibrillar morphometry of the non-treated corneas was calculated to be about 45% of the naïve corneal state.

### Cultivated keratocyte therapy improves the visual function of rats with acute corneal haze

To relate the corneal haze clearance with improvement in visual function, we subjected the naïve rats and rats that underwent cell therapies to visual-dependent behavioral tests in the Morris water maze (Fig. 5a) and tracked the rats' swim paths (Fig. 5b). The rats that received qCSK treatments had a similar performance to the naïve rats in terms of escape latency (Fig. 5c and Supplementary Table 16), swim distance from the start point to the escape platform (Fig. 5d), and swim path efficiency (Fig. 5e). The path efficiency indicated how direct the swim path is from the start point to the escape platform. The more direct the path is, the closer the efficiency to the value of 1. The qCSK treatment resulted in 1.47 ± 0.35x, 1.46 ± 0.28x, and 1.17 ± 0.27x improvement in escape latency, distance traveled, and path efficiency, respectively from the non-treated wounded rats.

The escape latency of the rats in SFs and non-treated groups was notably slower than the naïve rats ($p = 0.030$ SFs vs. naïve groups and $p = 0.036$ non-treated vs. naïve groups) (Fig. 5c and Supplementary Table 16). Despite disparities in distance traveled and path efficiency from the naïve group, the differences did not reach a significant level (Fig. 5d, e). The SF-injected animals performed with a rather similar efficiency but with 1.06 ± 0.13x slower escape latency and 1.10 ± 0.33x longer swim distance than the non-treated rats.

### Keratocyte therapy resolves chronic corneal opacity albeit with a marginally attenuated potency than in acute opacity

The excimer laser-induced haze model is a very 'clean' model of haze induction, as opposed to the others mentioned, such as chemical injury and microbial keratitis models, which often are dictated by the extent of vascularization[31,32]. In addition, more patients may present with such conditions in the clinic. Hence, to demonstrate the efficacy of the qCSK therapy further, we created a chronic opacity model via IrrPTK. Instead of administering cell therapy 7 days after the laser injury, we allowed the haze to establish further and performed the cell injection on day 28 (Fig. 6a). On the slit lamp, the established haze was typically accompanied by severe neovascularization (Fig. 6b). The qCSK therapy showed a steady reduction of haze and NV over 21 days (Fig. 6c, d and Supplementary Table 17). The SF injection did not alleviate the haze and NV, which outcomes resembled the state of the non-treated corneas (Fig. 6b, c). After 21 days, the total scores of the SFs and non-treated groups were markedly higher than the qCSKs group ($p = 0.001$ and $p = 0.001$, respectively) (Fig. 6e and Supplementary Table 17). Quantitatively, at PID21, the haze density after qCSK injection was significantly reduced compared to the SF injection ($p = 0.006$) and non-treated corneas ($p = 0.001$) (Fig. 6f and Supplementary Table 18). In the latter 2 groups, although the haze density decreased, the haze areas appeared to enlarge over time (Fig. 6g and Supplementary Table 19). The haze areas of both groups were greater than that after the qCSK injection at PID21 ($p = 0.001$ vs. SFs and $p = 0.068$ vs. non-treated). Unlike in the acute haze model, the CCT was close to the pre-operative value after 21 days in the qCSKs and non-treated groups (Fig. 6h and Supplementary Table 20). In contrast, the edema did not subside after SF injection, which resulted in 13.92 ± 7.97% thicker corneas than in the preoperative state.

Similar to the acute haze model, the therapeutic efficacy of qCSKs was significantly better than the SFs and non-treatment in the established haze model. However, the treatment potency was marginally lower in chronic opacity. For example, the median haze and NV scores were 1 (0.5 in the acute opacity model) and 2 (0.5 in the acute opacity model), respectively. Other parameters, such as haze density

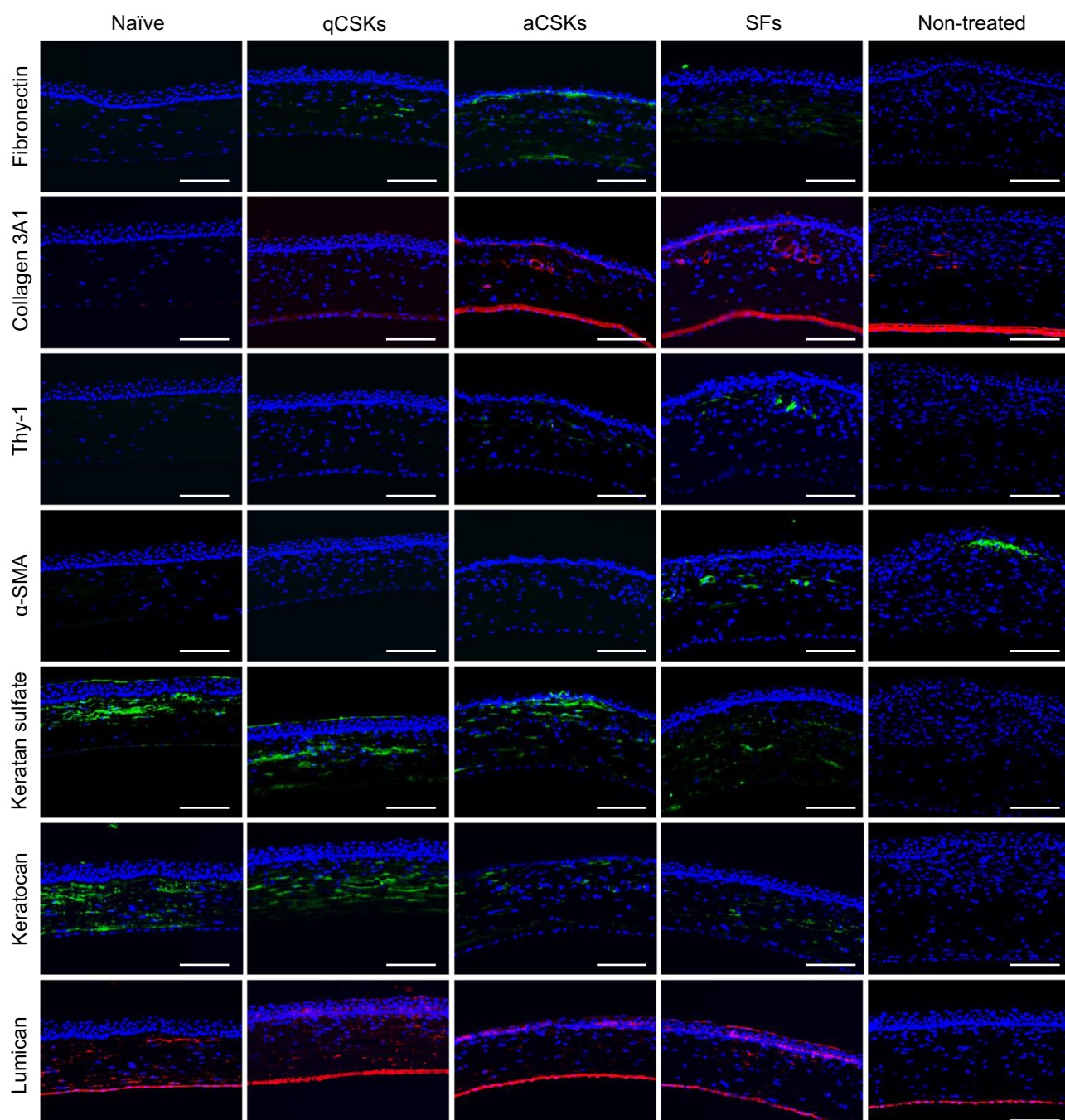

**Fig. 3 | Immunofluorescence staining of corneal fibrosis-associated proteins and keratan sulfate proteoglycans in laser-injured corneas undergoing cell therapies.** The corneas were harvested 21 days after cell injections. The quiescent corneal stromal keratocytes (qCSKs)-treated corneas exhibited little to no expression of Thy-1, α-smooth muscle actin (α-SMA), fibronectin, and collagen 3A1. The corneas that received activated CSKs (aCSKs) injections expressed Thy-1, α-SMA, fibronectin, and collagen 3A1 in the anterior stroma but at an attenuated level compared to the stromal fibroblasts (SFs)-injected corneas. The keratan sulfate (KS) and KS proteoglycans, keratocan and lumican, were largely absent in non-treated corneas but were re-expressed in the qCSK-treated corneal stroma close to the naïve corneal state. The aCSK- and SF-injected corneas also re-expressed the KS, keratocan, and lumican but their expression levels were lower than the naïve corneas and qCSK-treated corneas. Staining was repeated on 3 independent samples. Scale bars = 100 μm.

(9.93 ± 0.52 × 10⁶ arb.units vs. 8.19 ± 0.28 × 10⁶ arb.units in the acute opacity model) and haze area (28.37 ± 7.35% vs. 7.42 ± 1.09% in the acute opacity model) were also higher in the corneas with established haze.

## Discussion

The results indicate that with our methodology, qCSKs grown from culture can reliably replicate the function of native CSKs, following cell injection delivery. Of note, the KS proteoglycan expression of cultured qCSKs is more similar to native CSKs and can be readily distinguished from cultured SFs, adding to established evidence in the literature[3–5]. Our in vivo studies demonstrated that both the choice of cell types utilized for cell therapy (qCSKs, aCSKs, or SFs), as well as the quality of the cell injection procedure are critical parameters for the proper resolution of corneal haze. Specifically, qCSKs coupled with good quality delivery provided the most effective

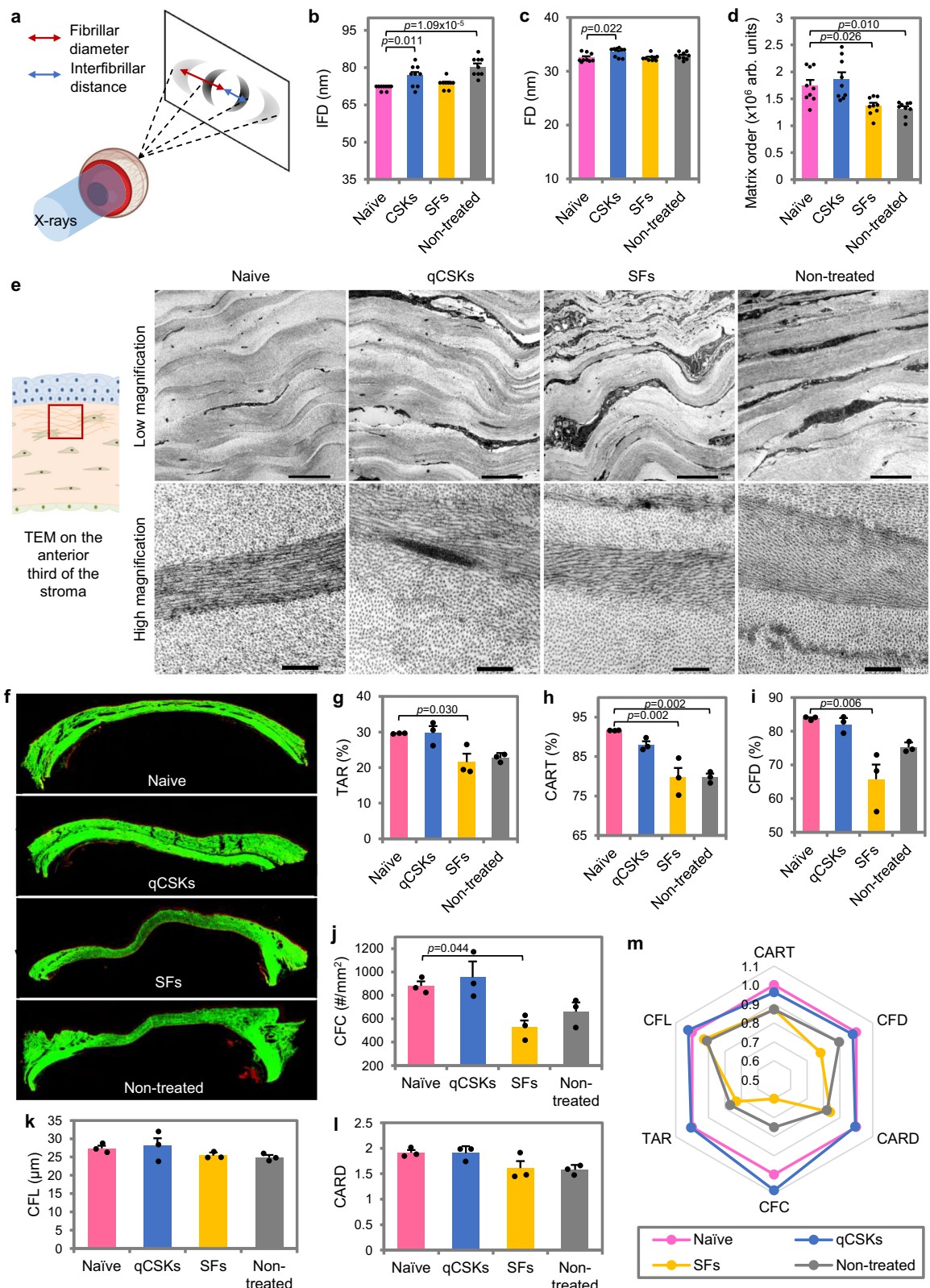

reduction of haze density and area. The outcome consistency of good quality qCSK delivery was demonstrated in two different instances. The 9 rats in Fig. S2 had a haze density and area of $8.70 \pm 0.27 \times 10^6$ arb.units and $11.55 \pm 2.70\%$, respectively at PID21. The results were not significantly different from the 6 rats in Fig. 2, which had a haze density and area of $8.19 \pm 0.28 \times 10^6$ arb. units ($p = 0.208$) and $7.42 \pm 1.09\%$ ($p = 0.185$), respectively. The poor

cellular delivery typically resulted in a lower and bigger range of volume expansion due to leakage or backflow of the qCSKs-containing injection medium, which appeared to correlate with the extent of haze clearance efficacy, which was poorer and highly variable in comparison to the good delivery outcomes. The experiment highlighted the importance of cell density and spatial localization in determining the therapeutic effects of the qCSK injection.

**Fig. 4 | Collagen fiber morphometry and organization in laser-ablated corneas undergoing cell therapies.** The corneas were harvested 21 days after cell injections. **a** Synchrotron small-angle X-ray scattering (SAXS) was employed to elucidate the interfibrillar distance (IFD), fibrillar diameter (FD), and degree of collagen fibril organization (matrix order) in the corneas centrally. The IFD (**b**) and FD (**c**) were only marginally different between all groups, namely the naïve, quiescent corneal stromal keratocytes (qCSKs)-treated, stromal fibroblasts (SFs)-treated, and non-treated corneas, with naïve corneas having the smallest IFD and FD. **d** The matrix order in the qCSK-injected corneas was akin to the naïve corneas and was significantly higher than the SFs-injected and non-treated samples. **e** Low-magnification transmission electron microscopy (TEM) images of the anterior stroma region revealed a more regular alternation of darker (collagen fibers that appeared in longitudinal orientation) and lighter (fibers that appeared in cross-sectional orientation) bands of lamellae in the naïve and qCSKs groups. At high magnification, especially within the darker band, the fibers were typically long and continuous in both naïve and qCSK-treated rats but appeared relatively shorter due to the interruption by the cross-sectionally appearing fibers in the lamellar layer. **f** Further collagen fiber profiles were characterized with a Histoindex Genesis 200 multiphoton microscope. The built-in analytical system then measured the tissue area ratio (TAR) (**g**), collagen area ratio in tissue (CART) (**h**), collagen fiber density (CFD) (**i**), collagen fiber count (CFC) (**j**), collagen fiber length (CFL) (**k**), and collagen area reticulation density (CARD) (**l**), revealing the restoration of collagen fiber profile by qCSKs to the naïve corneal state (**m**). In contrast, the collagen fiber profile of SF-injected corneas was closer to that of the non-treated corneas. Data are presented as mean ± SEM ($n = 9$ in each group for SAXS and $n = 3$ in each group for multiphoton microscopy). TEM was repeated on 3 independent samples. Statistical significance was analyzed with one-way ANOVA, followed by post hoc Tukey test. Naïve, qCSKs, SFs, and non-treated groups are represented by pink, blue, yellow, and gray, respectively, in the bar graphs and radar chart. Source data are provided as a Source Data file.

We also investigated the in vivo expression of KS proteoglycan and corneal-fibrosis-related proteins, as well as changes to the collagen microstructure within the native rat corneal milieu with respect to different cell types utilized for cell-injection therapy. It was demonstrated through immunofluorescence staining that injected qCSKs maintained their characteristic protein expression signatures in vivo and, unlike SFs, did not exhibit characteristics of myofibroblastic transformation. Injected qCSKs also expressed characteristic proteoglycans which resembled native cornea keratocytes as compared to SFs. SAXS, TEM, and multiphoton microscopy further confirmed that the injected qCSKs preserved their active role in remodeling and rearranging the collagen matrix towards a native configuration within the rat corneal stroma. This was demonstrated by the restoration of matrix order in qCSK cell injection, which when compared to SFs and non-treated controls, possessed a collagen architecture closer to the naïve state. For a functional visual study, we utilized a modified Morris Water Maze test as an analog to determine if cell injection therapy led to appreciable improvements in vision[33]. We demonstrated that compared to SF-treated and untreated wound rats, qCSK-treated rats had a significant improvement in escape latency and efficiency, further supporting qCSKs as a preferred cell type for cell injection therapy. Taken together, the ability to culture and expand qCSK with preserved intrinsic function, paired with optimized cell injection techniques, paves the road ahead for the viability of utilizing qCSKs for the treatment of corneal haze. The in vitro cell expansion has the potential to overcome critical supply limitations and the almost non-invasive stromal injection method would very likely have an improved efficacy and safety profile as compared to current surgical solutions.

Further exploring the nuances and complexity of the corneal haze and scar dynamics, evidence exists in the literature which shows that in response to corneal injury, native CSKs activate from quiescence and typically quickly lose their proteoglycan expression phenotypes and transform into proliferative SFs and myoSFs with distinct gene expression profiles[25,34], resulting in corneal opacification. This natural pathologic response to corneal injury has been demonstrated to result from exposure to inflammatory and proliferative cytokines, such as IL-1α, TGF-β1, and tumor necrosis factor-α (TNF-α)[9,35]. Bone-marrow-derived fibroblasts are also recruited into the site of injury, transforming into bone-marrow-derived myofibroblasts. MyoSFs and bone-marrow-derived myofibroblasts bring about corneal scarring and fibrosis by unorganized ECM deposition within the stroma[36]. In particular, the ECM increases in density with the deposition of collagen types III and IV, fibronectin, and tenascin-C[37]. It is important to note that the generation SFs and myoSFs are part of the normal wound healing process, which helps to ensure closure of the wound bed by secretion of adhesive ECM components and initiating wound contraction. The depth of the wound, as well as the pathogenesis of the injury (e.g., chemical, infectious, or trauma), dictate the severity of the disease and scar response[38]. Studies in photorefractive keratectomy (PRK) surgery and alkali burn injuries demonstrate that the depth of the induced wound and the type of injury led to a differential abundance of myofibroblasts. After the formation of a corneal haze or scar, fibrotic tissue is then gradually replaced by corneal stroma over a period of months to years. This is largely dependent on the regeneration of the basement membrane and reduction in active myofibroblasts[39]. It is currently not well understood how the exact process of scar resolution occurs within the cornea, it is likely to be a complex process involving immune modulation (inhibition of CD3+ T lymphocytes, VEGF, MMP-9, TNF-α, IL-1β, IL-6, IL-10, and PDGF), regeneration of basement membranes, and activation of resident keratocytes. Furthermore, knowledge of the exact cell-cell interactions that enable introduced qCSKs to modulate corneal haze and scar resolution is largely unknown.

Nevertheless, this study supported our hypothesis that cultured qCSKs maintain their intrinsic function after cell injection and can successfully resolve acute corneal haze in a rat model. Of note, there was a correlation between the degree of haze resolution and the degree of corneal stromal cell "differentiation" from qCSKs to aCSKs and to SFs. In vitro, we have been able to derive SFs from CSKs by increasing the serum concentration in growth media during cell expansion. We have also previously demonstrated the generation of qCSKs from aCSKs by serum deprivation in culture. We hypothesize that as cultured cells "differentiate" further from the native CSK state under proliferative pressures, their functional CSK role becomes diminished. It is known that in the corneal scarring pathology, under profibrotic differentiation pressures, native CSKs transform into SFs and myoSFs. MyoSFs negligibly express corneal crystallins and overexpress α-SMA, contributing to opacification and wound contraction, respectively[12]. It is likely that the differential effects of haze area score between qCSK vs. SF injection arise from the SFs' poorer recapitulation of native CSK function and qCSK possessing closer functional resemblance to native CSKs. One possible mechanism of haze remediation by qCSKs can be inferred from the difference in collagen fibril arrangement as seen by SAXS and TEM analysis. While SF injection resulted in a deviation of fibril matrix order from the native cornea, qCSKs did not. This suggests that SFs had a reduced production and deposition of appropriate KS proteoglycans as well as a reduced ability to retain physiologic collagen fibril homeostasis compared to qCSKs (Fig. 7). Furthermore, qCSKs, unlike SFs, are not known to induce pro-inflammatory or pro-fibrotic cytokines and chemokines[40,41]. In our rat model, the cytokines and chemokines secreted by the injected SFs most likely have a synergistic and compounding effect with myofibroblastic transformation and inflammatory responses within the injured corneas, resulting in persistent haze and increasingly severe NV. The

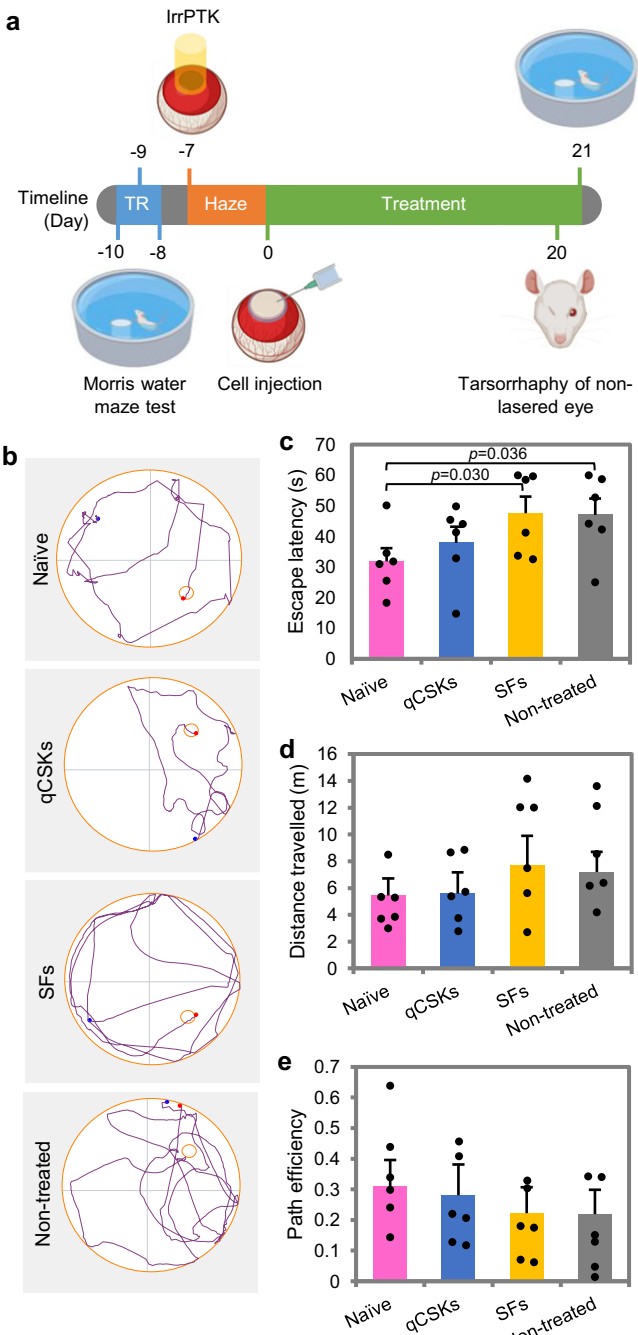

**Fig. 5 | Visual-dependent behavioral assessment of rats that underwent cell therapies with Morris water maze. a** Experimental timeline showing the swim training regime (TR), acute corneal haze induction with irregular phototherapeutic keratectomy (IrrPTK) and haze development, cell therapy, and closure of the eyelid of the non-lasered eye a day before the final test. **b** The swim path maps indicated that naïve and quiescent corneal stromal keratocytes (qCSKs)-treated rats performed better, e.g., traveled a shorter distance from the start point (black spots) to the escape platform (red spots) than the stromal fibroblasts (SFs) and non-treated groups. The qCSK-injected rats (blue bars) displayed similar escape latency (**c**), distance traveled (**d**), and path efficiency (**e**) as the naïve rats (pink bars). On the other hand, the SFs-treated (yellow bars) and non-treated (gray bars) rats had higher escape latency and distance traveled, and lower path efficiency than the naïve rats. Data are presented as mean ± SEM (*n* = 6 rats in each group, with each rat subjected to the trial thrice). Statistical significance was analyzed with one-way ANOVA, followed by post hoc Tukey test. Source data are provided as a Source Data file.

efficacy of qCSKs for haze resolution could be partly due to or improved by this lack of pro-inflammatory signaling.

We further investigated qCSK cell injection therapy in a chronic haze model to determine if the therapeutic temporal window could be extended beyond that of acute haze. The pool of patients affected by chronic haze worldwide is large, many of whom reside in medically under-served regions with trachoma as a leading etiology for the disease[2]. Cell therapy for the treatment of chronic haze would be an efficient use of an already scarce donor corneas. Our preliminary end-point result of partial scar resolution in a chronic haze model has been notably encouraging and further research is currently ongoing. It is important to note that the current research does not encompass the entire scope of challenges in chronic corneal haze and scarring. Chronic haze, as compared to acute haze, likely possesses a different microstructural makeup, cellular microenvironment, and immunological landscape as compared to acute scars. Empirically, in our chronic haze model, increased vascularization into the cornea, accompanied by dense stromal haze has been observed. Investigation of the exact pathophysiology of the evolution of a chronic scar remains outside the purview of this study, but we acknowledge that further research in these areas is essential for developing comprehensive therapies for chronic corneal haze and scarring. Two different approaches to tackling chronic haze will be explored in the future. First, because injected qCSKs have a half-life of 3 weeks, an additional injection of qCSKs after 3 weeks may be required to resolve the haze further. Because of the prominent involvement of the immunological component, it is also possible that the remaining scarred tissue is not repairable by the qCSKs. The second strategy, hence, involves the removal of scar tissue, which will then be replaced by a biomaterial, such as collagen or hyaluronic acid[42,43], encapsulated with qCSKs. The biomaterial would act as a temporary scaffold, permitting time for the qCSKs to secrete the ECM proper and organize the collagen fibers.

The value proposition of stromal cell-injection therapy with qCSKs will always be compared with its alternative: adult CSSCs, which is at a more advanced stage of translation into clinical therapeutics[44,45]. Although the CSSCs in the peripheral cornea and limbus can adopt CSK-like phenotypes in vitro[46], it remains to be seen whether the cells undergo preferred differentiation to CSKs in vivo due to elevated levels of pro-fibrotic TGF-β1 and TGF-β2 post-CSSC transplantation[47]. We chose to cultivate CSKs from the central corneal stroma as the stromal cell population in the central region is less heterogeneous and contains predominantly quiescent CSKs. CSKs have a lower propensity for fibroblastic transformation at low serum concentrations and are relatively easy to maintain and culture, all while possessing more efficient ECM remodeling capabilities than the CSSCs[41]. CSSCs, which—through current limitations in isolation techniques—are composed of a heterogeneous population of cells, including mesenchymal stem cells and neuronal precursor cells. Furthermore, while CSSCs have demonstrated superior immunomodulatory effects, this is likely to be the primary mechanism employed by CSSCs to mitigate corneal opacity[48], and unlike qCSKs, their contribution towards corneal stromal homeostasis and remodeling need further investigation.

In an effort to optimize qCSK production, we cultured donor-derived SFs in 10% serum, followed by a 14-day serum starvation in serum-free media, aiming to test if this was a viable alternative methodology, which owing to the increased proliferative capacity of SFs, could potentially increase qCSK yield. This method yielded cells with low levels of qCSK markers, such as ALDH1A1 and lumican but with markedly higher ALDH3A1 and keratocan. Due to the failure to replicate qCSK morphology and expression profiles fully, these serum-starved SFs were deemed unsuitable as qCSK alternatives. The transformation of isolated keratocytes in high serum conditions is likely to be an irreversible process once the SF phenotype is established. Expectedly, reversal of the SFs phenotype to qCSKs under serum-free conditions has been demonstrated here to be ineffective, although the

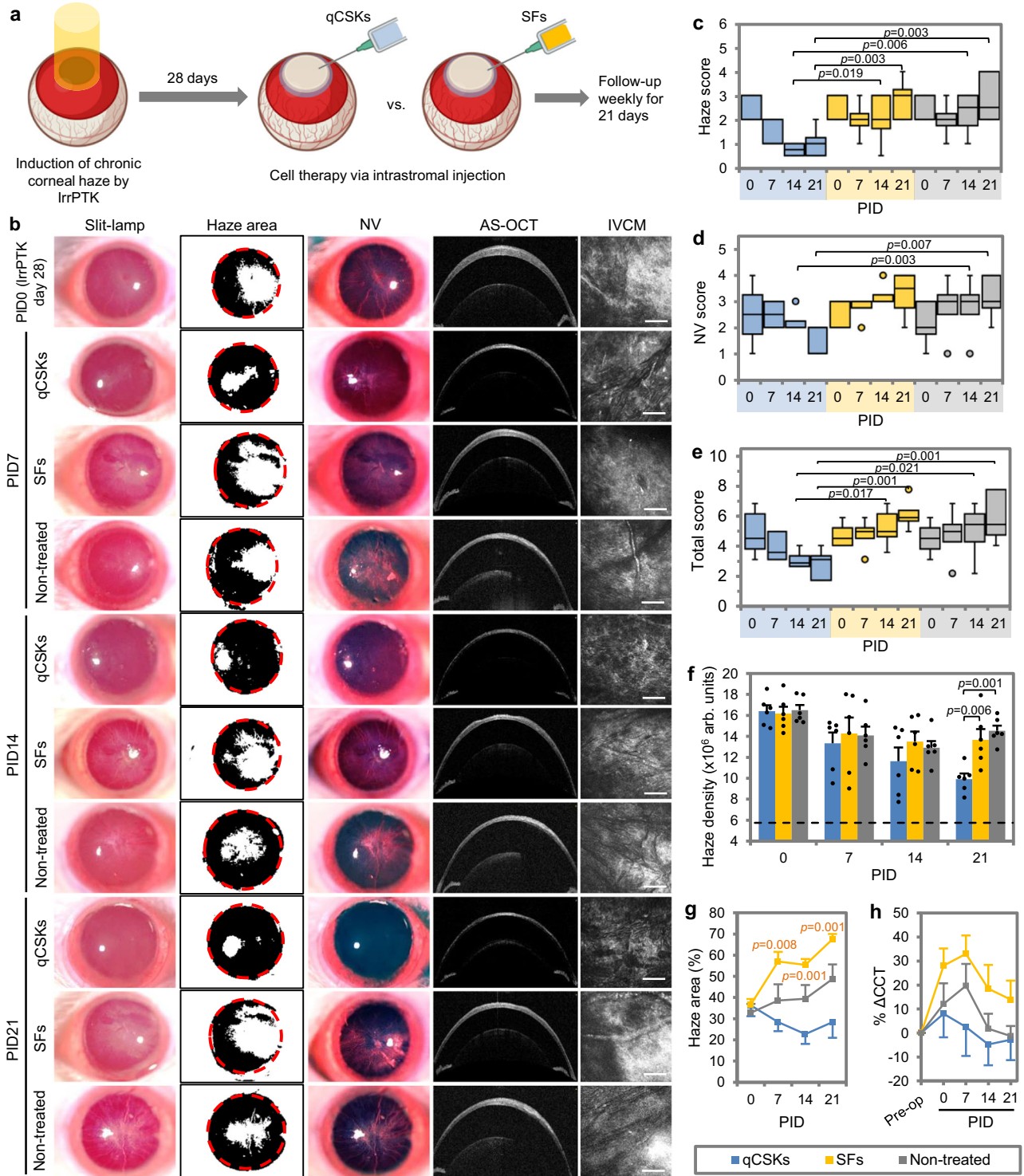

**Fig. 6 | Postoperative outcomes of cell injections in corneas with excimer laser-induced chronic opacity. a** Experimental setup and timeline. The quiescent corneal stromal keratocytes (qCSKs) and stromal fibroblasts (SFs) injection outcomes were compared to the non-treated corneas. **b** The corneal imaging showed that qCSKs had the best therapeutic efficiency. SF injection exacerbated the haze and neovascularization (NV) severity in a similar manner to the non-treated corneas. The slit-lamp observation concurred with the haze (**c**), NV (**d**), and total grades (**e**), and was further substantiated by the in vivo confocal microscopy (IVCM)-based haze density measurements (**f**). The dash line indicates the mean of haze density preoperatively. In the box plots, the center line shows the median score; box limits show the 1st and 3rd quartile; whiskers show minimum and maximum values; and points indicate outliers. Statistical significance was analyzed with Kruskal–Wallis,

followed by post hoc Dune-Bonferroni test. **g** The haze areas increased over time in the SF-injected and non-treated corneas. In contrast, treatment with qCSKs reduced the chronic haze area. The significant *p* values were relative to the qCSKs. **h** The central corneal thickness (CCT) returned to the preoperative state at a more rapid rate following qCSK injection compared to the non-treated group. The corneas remained thicker than the preoperated state even at 3 weeks after SF administration. Data are presented as mean ± SEM. Statistical significance was analyzed with one-way ANOVA, followed by post hoc Tukey test. The qCSKs, SFs, and non-treated are represented by blue, yellow, and gray in the box plots, bar graphs, and line graphs. *n* = 6 rats in each group. Scale bars = 100 μm. Source data are provided as a Source Data file.

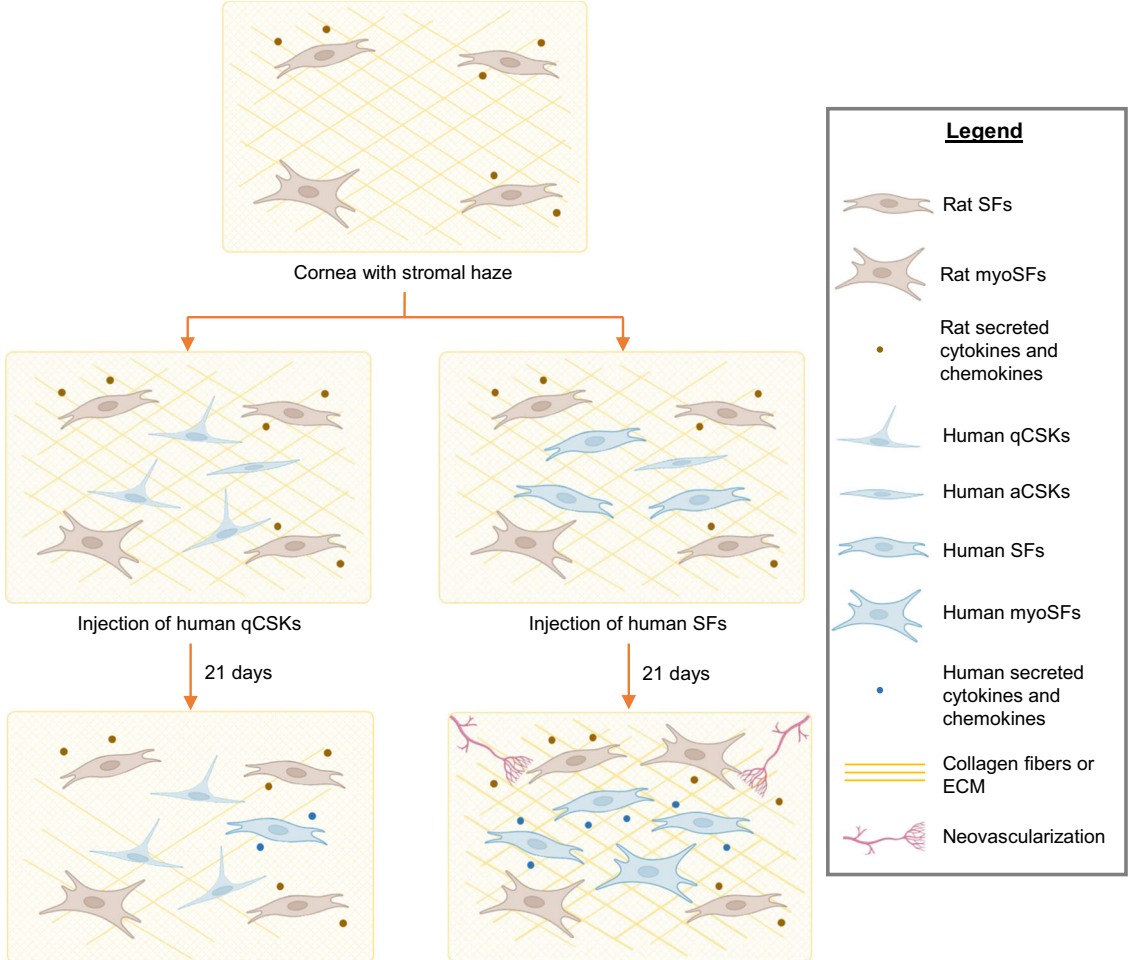

**Fig. 7 | Mechanism of haze clearance following quiescent corneal stromal keratocyte versus stromal fibroblast injections.** Following corneal injury, inflammatory and proliferative cytokines induced by the fibroblastic transformation of the native corneal stromal keratocytes (CSKs) result in abnormal extracellular matrix (ECM) or disorganized collagen fibers. Due to the relatively quiescent nature of injected quiescent CSKs (qCSKs), they are likely able to exert their intrinsic function to secrete the proper ECM and reorganize the collagen fibers. In contrast, the injected stromal fibroblasts (SFs) released cytokines/chemokines that compound the myofibroblastic transformation and inflammatory responses in the injured cornea. Cell injection of SFs likely results in the persistent presence of abnormal ECM, which in turn causes unabated corneal haze. Furthermore, the cytokines/chemokines naturally released by SFs likely contribute to increased neovascularization.

formation of an intermediary cell type with robust production of ALDH3A1 was surprising.

Nevertheless, our qCSK culture technique offers scalability, feasibility, and potential cost-effectiveness for clinical application. Our current protocol enables propagation of qCSKs, yielding a minimum of 2 million cells in a T75 flask at passage 4 per donor cornea, enough for over 12 treatments at a projected per injection cell count of 150,000 cells. This efficiency, coupled with the use of otherwise non-transplant grade tissue, enhances resource utilization and cost-effectiveness. The overall cost-effectiveness is a multifactorial consideration, encompassing the expenses related to the procurement of tissue, culture media, consumables, labor, regulation, and distribution. Given the nascent stage of our research within the rapidly evolving sector of Advanced Therapy Medicinal Products and cell therapy, a conclusive assessment of economic viability in clinical applications remains an estimate. However, given a similar capital outlay and raw material requirement as corneal endothelium cell therapy—a separate endeavor by our lab—and not accounting for synergism, similar cost estimations of $880 for cell therapy vs. $3710 for transplantation are likely[49]. Ongoing research is focused on refining cost evaluations, with preliminary comparisons suggesting significant cost advantages over conventional transplantation methods. Additionally, the study of utilizing post-cryopreserved qCSKs for cell therapy is ongoing alongside Good Manufacturing Practice conversion of materials and reagents.

A major limitation of this current study is that the results are derived from xenografts (cell injection of human cells within a rat corneal stroma), limiting generalizability to human therapeutics. Nevertheless, similar promising results have been demonstrated in other mammalian model organisms, and combined with this study[50], serve as a critical next step forward toward a clinical study. It is also important to note that cultured qCSKs do not mimic the native keratocytes entirely; we demonstrated that although cultured qCSKs are functional, their hallmark genetic markers are expressed at a lower level than the native CSKs and are in this sense more akin to SFs. There are therefore reasonable grounds to believe that further optimization of the culture protocol is required. Although it was empirically and statistically evident that the qCSK-injected rats performed at a similar level to the naïve rats, results of the modified Morris water maze test were unable to demonstrate significance in the total swim distance and path efficiency between qCSK-injected cohorts vs. SFs and non-treated study groups, except for the escape latency. A possible explanation for this deviation could be due to inherent methodological limitations or variances within the study

animals. To elaborate, the modified Morris water maze test is an analog for visual function derived from a test originally designed to measure rodent memory and behavior. Due to the interplay between neural systems and the visual sensorium, this method can be confounded by neurological differences, e.g., some rats learn more quickly. Large batch-to-batch variances also exist, i.e., some rats swim at higher speeds than others and although both groups of rats may escape at similar times, the distances that are covered are notably different. Nonetheless, there are few alternatives for a functional visual function test in rats and the current experimental setup serves to supplement the research question at hand[51].

Following the findings that there are some expression features that differ between native stroma and cultured qCSKs, further research is needed to establish if the culturing of qCSKs for intrastromal cell injection therapy is transcriptomically and genetically robust across passages, investigating this would help to validate if the phenotype observed can be maintained in culture. A follow-up study to track the gene expression changes within the scar resolution process would be of value and can provide clues as to how corneal opacity reduction by intrastromal cell injection therapies can be further targeted and optimized for a future therapeutic solution.

## Methods
The rat experiments were performed following The Association for Research in Vision and Ophthalmology (ARVO) Statement for the Use of Animals in Ophthalmic and Vision Research and approved by the Institutional Animal Care and Use Committee (IACUC) of SingHealth. The usage of anonymized human donor corneas did not require the approval of SingHealth Centralized Institutional Review Board because the ethics approval and collection of informed consent had been carried out by the U.S. eyebanks.

### Isolation and propagation of human corneal stromal cells
Fourteen pairs of research-grade cadaveric corneas deemed unsuitable for transplantation were procured from Lions World Vision Institute (Tampa, FL) and Saving Sight (Kansas City, MO) (Supplementary Table 21). The donor corneas were preserved in Optisol-GS (Bausch + Lomb, Bridgewater, NJ) and transported at 4 °C to the laboratory. The corneas were processed for cell culture within 24 h of receiving[52]. Briefly, the corneal stroma, after removing the epithelium, endothelium, and limbal stroma, was digested in 1 mg/ml of collagenase I (Worthington Biochemical Corporation, Lakewood, NJ), 0.1% bovine serum albumin (Sigma-Aldrich, St. Louis, MO) in CSK basal medium for 12 h at 37 °C. The basal medium consisted of DMEM/F-12 (Thermo Fisher Scientific, Waltham, MA), 0.1X insulin-transferrin-selenium (ITS; Thermo Fisher Scientific), 1X non-essential amino acids (NEAA; Thermo Fisher Scientific), 1X vitamin solution (Thermo Fisher Scientific), 1X antibiotic/antimycotic solution (Thermo Fisher Scientific), and 1X amino acids (Thermo Fisher Scientific). The isolated cells were collected for "aCSK" propagation using CSK full medium added with 0.5% fetal bovine serum (FBS; Thermo Fisher Scientific) on a collagen I-coated plate until passage 4. The CSK full medium was made of CSK basal medium, 5 μg/ml of human amniotic membrane extract, 10 ng/ml of insulin growth factor-1 (IGF-1; Thermo Fisher Scientific), 10 μM of ROCK inhibitor Y-27632 (Miltenyi Biotec, Bergisch Gladbach, Germany), and 0.5 mM of L-ascorbic 2-phosphate (Sigma-Aldrich). The aCSKs were passaged at ~70% confluency. To generate qCSKs, the aCSK culture was switched to a serum-free full medium for 14 days (Fig. 1a). To culture proliferative SFs, isolated stromal cells were placed in a 10% FBS-supplemented basal medium. To produce serum-starved SFs, the SF culture was switched to serum-free full medium for 14 days. All culture media were replaced every 3 days. All experiments were carried out using cells at passage 3 or 4.

### Gene expression analysis
Quantitative real-time polymerase chain reaction (RT-PCR) was performed on qCSKs, aCSKs, SFs, and human corneal stromal tissues from the same 3 donors. Briefly, the total RNA of cells or stromal tissues was extracted using a Qiagen RNeasy kit (Qiagen, Hilden, Germany) and purified using a Turbo DNA-free kit (Thermo Fisher Scientific). Reverse transcription of 1 μg of total RNA was performed using the SuperScript First-Strand Synthesis System (Thermo Fisher Scientific). Primers of CSK-related genes of interest were designed to accommodate the Roche Universal Probe Library system using the Roche ProbeFinder (Roche, Basel, Switzerland) (Supplementary Table 22). RT-PCR was carried out on a LightCycler 480 system (Roche) using a LightCycler 480 Probes Master master mix containing DNA polymerase (Roche). Samples were run in triplicates. The gene expression levels were normalized with the endogenous levels of *GAPDH*, and relative fold changes to the stromal tissues were analyzed using the $\Delta\Delta C_T$ method.

### Immunocytochemistry of keratocyte markers
The qCSKs, aCSKs, and SFs derived from the same 3 donors were seeded on collagen I-coated coverslips and fixed with 4% paraformaldehyde (Sigma-Aldrich), treated with ice-cold 50 mM ammonium chloride (Sigma-Aldrich), and permeabilized with 0.15% saponin (Sigma-Aldrich). The cells were then treated with 1.5 U of endo-β-galactosidase (Sigma-Aldrich) in 10 mM phosphate buffer, pH 7.4 for 30 min at 37 °C. After blocking with 1% bovine serum albumin (BSA; Sigma-Aldrich) and 2% normal goat serum (Thermo Fisher Scientific), the cells were incubated with rabbit polyclonal anti-keratocan (diluted 1:100; Atlas Antibodies, Broma, Sweden), anti-lumican (diluted 1:100; Bioss, Woburn, MA), and anti-ALDH3A1 (diluted 1:100; Proteintech, Rosemont, IL) antibodies for 2 h at room temperature. After a series of washes in 1x phosphate-buffered saline (PBS; 1st BASE, Singapore, Singapore), the cells were incubated in Red-X-conjugated secondary antibody (Jackson ImmunoResearch Laboratories, West Grove, PA) and AlexaFluor 488-conjugated phalloidin for 45 min and finally, mounted in Fluoroshield with 4,6-diamidino-2-phenylindole (DAPI; Santa Cruz Biotechnology, Dallas, TX) and viewed under an Axioplan 2 fluorescence microscope (Carl Zeiss, Oberkochen, Germany).

### Baseline effects and safety of cell injections
The baseline study was conducted on 18 naïve, non-injured Sprague-Dawley rats (male, 6–8 weeks old) in accordance with the IACUC of SingHealth (protocol no. 2019/SHS/1470). The rats were housed in conventional rat cages (2 rats/cage) (Techniplast, Buguggiate, Italy). The rats were fed with pellets (catalog no. 1324; Altromin, Lage, Germany). The cages were checked daily to ascertain that the rats had unlimited water and feed. The cages were placed in a room with a 12-h light-dark cycle at 18–26 °C and humidity of 55–75%. Before surgery, the rats were anesthetized with a combination of 80 mg/kg of ketamine (Parnell Laboratories, Alexandria, Australia) and 12 mg/kg of xylazine (Troy Laboratories, Glendenning, NSW, Australia) intraperitoneally. Following anesthesia, the rat eyes were administered topically with proparacaine (Alcaine; Alcon, Geneva, Switzerland). An experienced corneal surgeon (J.S.M.) then performed the intrastromal cell injection. A stromal tunnel at the edge of the haze region in the anterior stroma was first created with a 31G needle. Following that, $4 \times 10^4$ cells in 2 μl of 1x PBS (1st BASE) were injected through the tunnel, through a 30G blunt needle attached to a Hamilton syringe (Hamilton Company, Reno, NV). The rats were randomly placed into these treatment groups: qCSKs, aCSKs, and SFs groups. Thereafter, all eyes were instilled with tobramycin and dexamethasone (TobraDex; Alcon) 4 times daily for 7 days and examined weekly. The rats were euthanized 21 days post-injection.

**Table 1 | List of antibodies used to stain corneal tissue sections**

| Antibody (clone number) | Source | Manufacturer | Dilution factor |
|---|---|---|---|
| Thy-1 (OX-7) | Mouse monoclonal | Novus Biologicals (Centennial, CO) | 1:200 |
| Fibronectin (DH1) | Mouse monoclonal | Sigma-Aldrich (St. Louis, MO) | 1:200 |
| Keratan sulfate (5D4) | Mouse monoclonal | MyBioSource (San Diego, CA) | 1:100 |
| Keratocan | Rabbit polyclonal | MyBioSource | 1:100 |
| Lumican (JE11-45) | Rabbit monoclonal | HUABIO (Woburn, MA) | 1:100 |
| α-SMA (1A4) | Mouse monoclonal | Agilent Technologies (Santa Clara, CA) | 1:200 |
| Collagen 3A1 (1E7-D7/Col3) | Mouse monoclonal | Novus Biologicals | 1:200 |

α-SMA α-smooth muscle actin.

### Rat models of acute and chronic corneal opacities and delivery of cell therapy

The rat models of acute and chronic corneal haze were created on 129 Sprague-Dawley rats (male, 6–8 weeks old) in accordance with the IACUC of SingHealth (protocol no. 2019/SHS/1470). The rats were housed in conventional rat cages (2 rats/cage) (Techniplast). The rats were fed with pellets (catalog no. 1324; Altromin). The cages were checked daily to ascertain that the rats had unlimited water and feed. The cages were placed in a room with a 12-h light-dark cycle at 18–26 °C and humidity of 55–75%. Before surgery, the rats were anesthetized with a combination of ketamine (Parnell Laboratories) and xylazine (Troy Laboratories) intraperitoneally. Corneal haze was induced by IrrPTK on a random rat eye[53]. The non-lasered eye was used as naïve group. The corneal epithelium was first debrided with a #64 surgical blade (BD, Franklin Lakes, NJ) sparing the limbus. To create acute haze, the IrrPTK was then initiated within a 3-mm ablation zone on the central cornea and an ablation depth of 15 μm using a Technolas 217z excimer laser (Bausch + Lomb). Irregular stromal damage was induced by placing a fine metal mesh over the laser ablation area after firing 50% of the pulses. The rats received topical tobramycin (Tobrex; Alcon) 4 times daily for 3 days. Stromal cell injection was administered 7 days after the IrrPTK when mild corneal haze typically started to develop. To create chronic or established haze, 30 μm of the stroma was ablated by IrrPTK, and the injury was left untreated for 28 days. On the cell injection day, following anesthesia, the rat eyes were administered topically with proparacaine (Alcaine; Alcon). An experienced corneal surgeon (J.S.M.) performed the intrastromal cell injection. Before injection, a stromal tunnel at the edge of the haze region in the anterior stroma was created with a 31G needle. Following that, $4 \times 10^4$ cells in 2 μl of 1x PBS (1st BASE) were injected through the tunnel, through a 30G blunt needle attached to a Hamilton syringe (Hamilton Company). The rats were randomly placed into these treatment groups: qCSKs, aCSKs, SFs, or non-treated groups. Thereafter, all eyes were instilled with tobramycin and dexamethasone (TobraDex; Alcon) 4 times daily for 7 days in the acute haze model or for 21 days in the chronic haze model. The rats were euthanized 21 days post-injection.

### Postoperative corneal examination and analysis

All corneal imaging was performed 3 days before injection (preoperative examination), on the 7th, 14th, and 21st day after cell injection. Slit-lamp photographs were taken using a Zoom Slit Lamp NS-2D (Righton, Tokyo, Japan). The corneas, manifesting haze and neovascularization after cell injection, were graded according to Fantes et al. for haze and LeBlanc et al. for neovascularization by providing a masked grader (A.K.R.) with the high-resolution slit-lamp images[54,55]. In addition to unprocessed images, the slit-lamp images, converted to black and white to aid visualization of the haze. Separately, the red color of the images was enhanced to aid the visualization of blood vessels in the corneas by setting the minimum and maximal red saturation channel to 90 and 140, respectively during color balancing using ImageJ software version 1.53t (National Institutes of Health, Bethesda, MD). The purpose of the enhancement was solely to augment the visibility of blood vessels for readers, addressing the challenge of discerning these details in non-enhanced slit-lamp photographs, particularly when viewed in printed format.

Corneal cross-section visualization and measurement of CCT were performed using an Optovue anterior segment-optical coherence tomography (AS-OCT; Visionix, North Lombard, IL). Mean CCT was measured as the average of three measurements taken at the corneal center (0 mm) and 0.5 mm on either side of the center. Finally, en-face images of the corneas were captured with an HRT3 IVCM with Rostock corneal module (Heidelberg Engineering GmbH, Heidelberg, Germany). A carbomer gel (Vidisic; Mann Pharma, Berlin, Germany) was applied on the corneal surface as an immersion fluid. All corneas were examined centrally from the epithelium to the endothelium. The light reflectivity level at the center of the haze region in the anterior stroma was obtained by measuring the Integrated Density of the IVCM images using ImageJ software version 1.53t (National Institutes of Health).

### Immunofluorescence staining of rat cornea

The excised whole rat corneas ($n = 3$ in each experimental group) were fixed in 4% paraformaldehyde (Sigma-Aldrich) and embedded in an optimum cutting temperature compound (Leica Microsystems, Wetzlar, Germany) for cryosectioning at a thickness of 8 μm. Tissue sections were treated with ice-cold 50 mM ammonium chloride (Sigma-Aldrich), saponin permeabilized, and blocked with 2% BSA (Sigma-Aldrich) and 5% normal goat serum (Thermo Fisher Scientific), followed by incubation with primary antibodies (Table 1) overnight at 4 °C. For keratocan and lumican immunostaining, sections were pretreated with 1.5 U/ml of endo-β-galactosidase (Sigma-Aldrich) in 10 mM phosphate buffer, pH 7.4 for 30 min at 37 °C before blocking and antibody incubations. After a series of washes with 1x PBS (1st BASE), the sections were labeled with Red-X- or AlexaFluor 488-conjugated IgG secondary antibodies (Jackson ImmunoResearch Labs) for 1 h at room temperature. The sections were mounted with Fluoroshield with DAPI (Santa Cruz Biotechnology) and then viewed under a Zeiss Axioplan 2 microscope (Carl Zeiss).

### X-ray scattering analysis

The rat corneas ($n = 9$ in each group) were harvested, wrapped in Saran Wrap (SC Johnson, Racine, WI), snap-frozen in isopentane (Sigma-Aldrich), and stored at −80 °C until imaging. SAXS was carried out on Beamline I22 at the Diamond Light Source UK national synchrotron (Didcot, UK), using an X-ray beam with wavelength 0.1 nm and cross-section measuring 0.25 mm (horizontal) × 0.25 mm (vertical) at the specimen. Before X-ray exposure, each specimen was thawed, then carefully examined under a light microscope, and any displaying evidence of major tissue wrinkles or folds were excluded. For data collection, the cornea was placed, superior side uppermost, into a sealed Perspex (Lucite Group Ltd., Southampton, UK) chamber with a Mylar window (DuPont-Teijin, Middlesbrough, UK). Previous SAXS work had shown that the hydration of corneas examined in this way did not significantly change, even after several minutes of X-ray exposure[56]. Specimen alignment was achieved by an initial exposure of X-ray-sensitive

film placed in the specimen holder to locate the incident beam position. SAXS patterns, each resulting from an X-ray exposure of 1 s, were collected at 0.5-mm (horizontal) × 0.5-mm (vertical) intervals covering the whole cornea and were recorded electronically on an X-ray detector placed 4.15 m behind the specimen position. An in-house software developed in MATLAB R2016b (The MathWorks Inc., Natick, MA, USA) was used to determine the average interfibrillar distance, collagen fibril diameter, and matrix order from individual corneal SAXS patterns[57].

## Transmission electron microscopy

The rat corneas were cut into halves: One half was subjected to TEM and the other half to multiphoton microscopy. The corneas ($n = 3$ in each experimental group) were fixed in 2.5% glutaraldehyde (Sigma-Aldrich) in 0.1 M sodium cacodylate buffer (Electron Microscopy Sciences, Hatfield, PA) at 4 °C overnight. The tissues were then washed and trimmed into smaller pieces, post-fixed in 1% osmium tetroxide and potassium ferrocyanide (Electron Microscopy Sciences), dehydrated in increasing concentrations of ethanol, and embedded in Araldite (Electron Microscopy Sciences). Ultrathin sections of 80 nm were cut with a Leica EM UC7 ultramicrotome (Leica Microsystems, Wetzlar, Germany) and collected on copper grids, double stained with uranyl acetate and lead citrate for 8 min each, and then viewed on a JEM-1220 microscope (JEOL, Tokyo, Japan).

## Multiphoton image acquisition system

The excised rat corneas ($n = 3$ in each experimental group) were fixed in 4% paraformaldehyde (Sigma-Aldrich) and embedded in paraffin blocks. Serial tissue sections with a thickness of 5 μm were cut with a HistoCORE Biocut microtome (Leica Microsystems). Before imaging, the paraffin was removed with Histo-Clear (Electron Microscopy Sciences), and the tissues were rehydrated through changes of 100% and 70% ethanol. Images of unstained cryosections were acquired on a fully automated, programmable, multiphoton imaging platform (Genesis 200; HistoIndex, Singapore). Laser excitation was carried out at 780 nm, and forward-scatter two-photon excitation (TPE) and SHG signals were detected simultaneously using dedicated photomultiplier tubes for each channel at 550 and 390 nm, respectively, and by using a dichroic mirror (450DCLP, Omega) to separate TPE from SHG. Images were acquired at 20x with 512 × 512 pixel resolution and a dimension of 200 × 200 μm². TPE sensitivity was set to 0.75, and SHG sensitivity to 0.65. A bandpass filter with a center wavelength of 550 nm and bandwidth of 88 nm was set in front of the TPE photodetector. The laser baseline power was set at 0.45. The laser beam with horizontal polarization was used for excitation. The corneal collagen morphometric analysis was performed on FibroIndex software (HistoIndex).

## Vision-dependent behavioral testing

The Morris water maze test was used here with modifications[58]. The rats ($n = 6$ in each experimental group) were initially trained in a 1.2-m-diameter, 25 ± 2 °C water tank that was positioned in a well-lit room. A black escape platform (10.5 cm in diameter) that extended 1.5 cm above the water level was used. Black curtains were positioned around the pool to limit distal cues. Training consisted of three trials per day for 3 consecutive days. Intertrial intervals were approximately 15 min. During each trial, the rats were placed in the water from one of four equally spaced start locations (N, S, E, or W). The subjects were given up to 60 s to escape during each trial. If it did not escape within the allotted time, it was gently guided to the platform. For tracking the swimming duration, swimming distance, and escape path efficiency, ANY-Maze video tracking software (Stoelting Co., Wood Dale, IL) was used. Platform and start locations were randomized across each trial. The rats were towel-dried and transferred to their home cages under warm air between trials.

One day following the training, IrrPTK was performed, and cell injections were administered 7 days later. The final test consisted of 3 trials to assess the post-treatment visual function was conducted

21 days after cell injection. A day before the trial, the non-treated fellow eye of the rat was sutured closed. A naïve rat, used as a baseline, received the same sequence of training, eyelid tarsorrhaphy, and probe trial. The subjects were given up to 60 s to escape. If it did not escape within the allotted time, the escape time was recorded as this maximum. Distance traveled and swim path efficiency by the rats were also recorded. For these parameters, those that did not escape after the 60 s were excluded from the report due to the closer association of the parameters with animals' behavior than visual function, inducing extreme variability in the outcomes[51].

## Statistical analysis

All values were reported as mean ± standard error of the mean, except the clinical grading outcomes. Clinical scores were expressed in the median and interquartile range. The grading outcome comparisons were assessed with Kruskal–Wallis $H$-test, followed by post hoc Dunn-Bonferroni tests on SPSS version 27.0 (IBM, Armonk, NY). Otherwise, the statistical comparisons between groups were analyzed with one-way ANOVA, followed by post hoc Tukey tests or if between 2 groups, 2-tailed Mann–Whitney $U$ tests. A $p$ value lower than 0.05 was considered to be statistically significant.

## Reporting summary

Further information on research design is available in the Nature Portfolio Reporting Summary linked to this article.

## Data availability

The entire raw dataset of animal-related experiments is more than 4 TB in size and can be shared on request with appropriate data-transfer methods. The request can be made to the article's first (andri.kartasasmita.riau@seri.com.sg) or last author (jodhbir.s.mehta@singhealth.com.sg) with up to 7 days of time of response. Source data are provided with this paper.

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

## Acknowledgements

Illustrations of rat eyes, Morris water maze, cells, and blood vessels in Figs. 2, 4–7 and Supplementary Figs. 2–4 were created with BioRender.com (license no. EV26POC55V). The authors would like to thank Dr. Veluchamy Barathi's Ocular Translational Pre-Clinical Model Research Platform team at SERI for their assistance in the animal experiments, Ms. Jenetta Soo for the assistance in expanding the stromal cells, and grant support from the Singapore NMRC Clinician Scientist Award-Senior Investigator category (MOH-000197-00) that was awarded to J.S.M. and SingHealth Duke-NUS Academic Medicine Research Grant (AM/RM008/2024) awarded to A.K.R.

## Author contributions

A.K.R. and Z.L. wrote the paper. A.K.R., Z.L., G.H.Y., C.B., Q.M., E.J.H., N.Z.Y., H.S.O., T.-W.G., N.S.H., and J.S.M. performed the experiments. A.K.R., Z.L., C.B., Q.M., E.J.H., and N.Z.Y. analyzed and interpreted the data. A.K.R., G.H.Y., C.B., and J.S.M. supervised the experiments. A.K.R. and J.S.M. obtained funding. G.H.Y. and J.S.M. conceptualized and conceived the study.

## Competing interests

The authors declare no competing interests.
