## [Peer Review File · Nature Communications]

REVIEWER COMMENTS

Reviewer #1 (Remarks to the Author):

The authors in this study highlight the use of injection of serum-depleted, quiescent corneal stromal keratocytes (qCSKs) that were propagated from non-limbal stromal cells of human donor corneas via an “activated” state (aCSKs). The therapeutic efficacy of the qCSKs in a rat model of acute corneal opacity was evaluated which was determined on the basis of efficacy of cell delivery, which entails a single injection entry point, concentration of cells at the injury site, and choice of cell population type, which drives the corneal opacity resolution.

Overall a useful and interesting study with great potential in the future. The study is especially useful as there is limited work in categorically defining the quality of cell injection delivery techniques, as well as how on the effect of the quality depends on the cell types chosen.

Although the authors state that the “good” quality delivery provided the most effective reduction of haze density and area in animal model; how consistent and reproducible was this finding.

Although in this paper the haze was produced after excimer laser, the response of corneal haze subsequent to infection, immunologic process, trauma , chemical injury etc could be different . This would also be dictated by the extent and depth of vascularisation.

Further the manuscript can be shortened and made more concise as there is repetition at few places

Reviewer #2 (Remarks to the Author):

This is a nice study by the authors on cell therapy for corneal scarring. The authors offer interesting findings but in its current form the study may not quite be ready for publication. Here are a few comments that may be helpful

1. In terms of significance of the findings, it would obviously be much more valuable if the study had focused on an established scar rather than a developing scar.
2. How reproducible is the "irregular" PTK model in their hands in terms of the degree of opacity?
3. Regarding lines 843-844, enhancing image colors, particularly red, might give a misleading representation. A better approach would be to use high-quality images without enhancements, or if enhancement is necessary, to provide a detailed method of enhancement.
4. Can the authors comment on feasibility and cost-effectiveness relative to other cell therapy techniques? Can keratocyte culture be scaled up for clinical application?
5. Were the cells injected post-preparation without any freezing? Is this a practical approach for clinical application?
6. The injection of 4×10^4 cells in 2 μ l of 1x PBS leads to a large error, hence any minor loss during injection, evident from the provided videos, could impact the results. Did they consider using a 5 μ l volume instead?
7. Was there an attempt to render the SFs quiescent? In other words, what if SFs are also kept in serum free media for 10 days before administration. An exploration of this could deepen the study's scope.
8. Clarification is needed about the injection site concerning the PTK injury site. Was the injured area targeted, or was it the surrounding region?
9. The percentage of "unsuccessful" deliveries, as depicted in Fig 2, should be elucidated. Out of the 18 rats, how many had unsuccessful deliveries?
10. Including unsuccessful deliveries in statistical evaluations may not serve the paper's intent. Given their inherent variability from multiple tries for injection to perforation, I recommend their exclusion from the study.
11. The authors note, "We established a metric to categorize the efficacy of cell injections as either 'good' or 'poor' procedures based on several observable visual indicators. Our findings indicated that both the quality of cell delivery and the choice of cell type play pivotal roles in resolving corneal haze and restoring visual function, with qCSKs in conjunction with 'good' quality injections yielding the most favorable outcomes." (lines 156-160). However, given the attached videos which depict injections with varying cell volumes, including those of poor quality might not be conducive to the central premise of the study especially concerning the limited number of 18 rats. In general, I would recommend a large number of animals.
12. Considering that corneal regeneration in rats is slower than in mice, a longer follow-up duration than 21 days is advisable.
13. Lines 231-234 detail that a "good" delivery quality is determined by: a single injection entry point and the formation of a cell-containing bleb at the cornea's center, which occupies >17% of the corneal area, typically translating to a 1.8x-4.8x volume expansion. However, in other sections, it's mentioned that a 2 μ l injection of cells in PBS is the norm. This creates ambiguity regarding whether a successful injection means exactly 2 μ l or if it's about the resulting expansion (>17% of the corneal area). If these

two parameters are equivalent, their correlation should be elucidated. I'd suggest standardizing to intrastromal injections and excluding deviations from this protocol in the study's dataset.

14. A viability test is warranted to assess cell survival after passage through a 30-gauge needle.

15. Lastly, results like the one regarding the adverse outcomes of "poor" delivery (e.g., increased neovascularization) don't substantively advance the study's narrative due to the vast variability in the "poor delivery" category. Like this "Poor" delivery also resulted in increasing neovascularization (NV) severity scores over time (Fig. 2b and 2d and Table S2). "

8 January 2024

Dear reviewers,

Thank you for allowing us to submit a revised draft of NCOMMS-23-27906 for publication in *Nature Communications*. We express our sincere appreciation to you for dedicating time and effort to provide insightful comments and constructive feedback on our manuscript. In response to these valuable inputs, we have thoroughly revised the manuscript, incorporating most of the suggestions made by the reviewers. The recommendations from the reviewers and our corresponding responses are listed in this letter, ensuring that all changes are tracked for easy reference. Below, we present our point-by-point responses to each comment, as well as excerpts from the revised manuscript in blue, illustrating how they have been addressed in the revised manuscript. All page and line numbers refer to the revised manuscript with tracked changes (download the Word document file).

Reviewer #1:

The authors in this study highlight the use of injection of serum-depleted, quiescent corneal stromal keratocytes (qCSKs) that were propagated from non-limbal stromal cells of human donor corneas via an “activated” state (aCSKs). The therapeutic efficacy of the qCSKs in a rat model of acute corneal opacity was evaluated which was determined on the basis of efficacy of cell delivery, which entails a single injection entry point, concentration of cells at the injury site, and choice of cell population type, which drives the corneal opacity resolution.

Overall a useful and interesting study with great potential in the future. The study is especially useful as there is limited work in categorically defining the quality of cell injection delivery techniques, as well as how the effect of the quality depends on the cell types chosen.

Author Response: We thank the reviewer for the compliment.

1. Although the authors state that the “good” quality delivery provided the most effective reduction of haze density and area in animal model; how consistent and reproducible was this finding.

Author Response: The consistency and reproducibility of the “good” qCSK delivery outcomes were demonstrated by comparing the extent of haze density and area reduction on post-injection day (PID) 21 in Fig. S2 and Fig. 2. The two figures represented two experiments performed on different days. We have added the following statement in the

Discussion (page 31, lines 645-649) in response to the reviewer's comment: "The outcome consistency of "good" quality qCSK delivery was demonstrated in two different instances. The 9 rats in Fig. S2 had a haze density and area of $8.70 \pm 0.27 \times 10^6$ a.u. and $11.55 \pm 2.70\%$, respectively at PID21. The results were not significantly different than the 6 rats in Fig. 2, which had a haze density and area of $8.19 \pm 0.28 \times 10^6$ a.u. ($p=0.208$) and $7.42 \pm 1.09\%$ ($p=0.185$), respectively."

2. Although in this paper the haze was produced after excimer laser, the response of corneal haze subsequent to infection, immunologic process, trauma, chemical injury etc could be different. This would also be dictated by the extent and depth of vascularisation.

Author Response: Thank you, we agree with the reviewer's comment. The excimer-induced model is a very 'clean' model of haze induction, as opposed to the others mentioned. In the haze responses mentioned by the reviewer, the therapy would typically be required following the acute insult. Hence, to further demonstrate the efficacy, we created a chronic opacity model to further show the efficacy of the qCSK injection. The differences in the chronic opacity model were the ablation depth (30 μm vs. 15 μm in the acute haze model) and the time of therapeutic intervention (28 days vs. 7 days after haze induction with excimer laser). In summary, we demonstrated that the qCSK therapy resolved chronic corneal opacity albeit with an attenuated potency than in acute opacity. The following section has been added to the Results (pages 28-29, lines 586-616) and is copied inline below:

"The excimer laser-induced haze model is a very 'clean' model of haze induction, as opposed to the others mentioned, such as chemical injury and microbial keratitis models, which often are dictated by the extent of vascularisation.^{31,32} In addition, more patients may present with such conditions in the clinic. Hence, to demonstrate the efficacy of the qCSK therapy further, we created a chronic opacity model via IrrPTK. Instead of administering cell therapy 7 days after the laser injury, we allowed the haze to establish further and performed the cell injection on day 28 (**Fig. 6a**). On the slit lamp, the established haze was typically accompanied by severe neovascularization (**Fig. 6b**). The qCSK therapy showed a steady reduction of haze and NV over 21 days (**Fig. 6c** and **6d** and **Table S17**). The SF injection did not alleviate the haze and NV, which outcomes resembled the state of the non-treated corneas (**Fig. 6b** and **6c**). After 21 days, the total scores of the SFs and non-treated groups were markedly higher than the qCSKs group ($p=0.001$ and $p=0.001$, respectively) (**Fig. 6e** and **Table S17**). Quantitatively, at PID21, the haze density after qCSK injection was significantly reduced compared to the SF injection ($p=0.006$) and non-treated corneas ($p=0.001$) (**Fig. 6f** and **Table S18**). In the latter 2 groups, although the

haze density decreased, the haze areas appeared to enlarge over time (**Fig. 6g** and **Table S19**). The haze areas of both groups were greater than that after the qCSK injection at PID21 ($p=0.001$ vs. SFs and $p=0.068$ vs. non-treated). Unlike in the acute haze model, the CCT was close to the pre-operative value after 21 days in the qCSKs and non-treated groups (**Fig. 6h** and **Table S20**). In contrast, the edema did not subside after SF injection, which resulted in $13.92\pm 7.97\%$ thicker corneas than in the preoperative state.

Similar to the acute haze model, the therapeutic efficacy of qCSKs was significantly better than the SFs and non-treatment in the established haze model. However, the treatment potency was marginally lower in chronic opacity. For example, the median haze and NV scores were 1 (0.5 in the acute opacity model) and 2 (0.5 in the acute opacity model), respectively. Other parameters, such as haze density ($9.93\pm 0.52 \times 10^6$ a.u. vs. $8.19\pm 0.28 \times 10^6$ a.u. in the acute opacity model) and haze area ($28.37\pm 7.35\%$ vs. $7.42\pm 1.09\%$ in the acute opacity model) were also higher in the corneas with established haze.”

The following discussion on the outcome of the chronic opacity model was added to the Discussion (page 37, lines 751-775): “We further investigated qCSK cell injection therapy in a chronic haze model to determine if the therapeutic temporal window could be extended beyond that of acute haze. The pool of patients affected by chronic haze worldwide is large, many of whom reside in medically under-served regions with trachoma as a leading etiology for the disease.² Cell therapy for the treatment of chronic haze would be an efficient use of an already scarce (donor corneas). Our preliminary end-point result of partial scar resolution in a chronic haze model has been notably encouraging and further research is currently ongoing. It is important to note that the current research does not encompass the entire scope of challenges in chronic corneal haze and scarring. Chronic haze, as compared to acute haze, likely possesses a different microstructural makeup, cellular microenvironment, and immunological landscape as compared to acute scars. Empirically, in our chronic haze model, increased vascularization into the cornea, accompanied by dense stromal haze has been observed. Investigation of the exact pathophysiology of the evolution of a chronic scar remains outside the purview of this study, but we acknowledge that further research in these areas is essential for developing comprehensive therapies for chronic corneal haze and scarring. Two different approaches to tackling chronic haze will be explored in the future. First, because injected qCSKs have a half-life of 3 weeks, an additional injection of qCSKs after 3 weeks may be required to resolve the haze further. Because of the prominent involvement of the immunological component, it is also possible that the remaining scarred tissue is not repairable by the qCSKs. The second strategy, hence, involves the removal of scar tissue, which will then

be replaced by a biomaterial, such as collagen or hyaluronic acid,^{42,43} encapsulated with qCSKs. The biomaterial would act as a temporary scaffold, permitting time for the qCSKs to secrete the ECM proper and organize the collagen fibers.”

3. Further the manuscript can be shortened and made more concise as there is repetition at few places.

Author Response: The manuscript has been shortened by moving two sections from pages 12-14, lines 276-312 and pages 14-15, lines 324-347 in the Results to the Supplementary Results in the Supplementary Information (pages 2-4). The moved sections were about the detailed description of results in Fig. S2 and S3.

Reviewer #2:

This is a nice study by the authors on cell therapy for corneal scarring. The authors offer interesting findings but in its current form the study may not quite be ready for publication. Here are a few comments that may be helpful.

Author Response: We thank the reviewer for the compliment and appreciate the reviewer’s feedback. We have performed additional experiments and improved the quality of the manuscript. Point-by-point replies to the reviewer’s specific queries are also included below.

1. In terms of significance of the findings, it would obviously be much more valuable if the study had focused on an established scar rather than a developing scar.

Author Response: Thank you for highlighting the importance of focusing on established scars, as well as developing scars. Acknowledging this, we have conducted additional experiments to assess the effectiveness of the cultivated quiescent corneal stromal keratocytes (qCSKs) on established corneal haze. This new experiment utilized irregular phototherapeutic keratectomy (IrrPTK) for injury induction. To model established haze, we ablated a depth of 30 μm from the stroma, in contrast to the 15 μm used for creating an acute haze model, and allowed 28 days for haze development, compared to the 7 days for acute haze. A notable distinction in the chronic haze model is the marked increase in blood vessel invasion towards the central cornea and a greater density of haze, differentiating it from the acute haze model. The following section has been added to the Results (pages 28-29, lines 586-616) and is copied below:

“The excimer laser-induced haze model is a very ‘clean’ model of haze induction, as opposed to the others mentioned, such as chemical injury and microbial keratitis models,

which often are dictated by the extent of vascularisation.^{31,32} In addition, more patients may present with such conditions in the clinic. Hence, to demonstrate the efficacy of the qCSK therapy further, we created a chronic opacity model via IrrPTK. Instead of administering cell therapy 7 days after the laser injury, we allowed the haze to establish further and performed the cell injection on day 28 (**Fig. 6a**). On the slit lamp, the established haze was typically accompanied by severe neovascularization (**Fig. 6b**). The qCSK therapy showed a steady reduction of haze and NV over 21 days (**Fig. 6c** and **6d** and **Table S17**). The SF injection did not alleviate the haze and NV, which outcomes resembled the state of the non-treated corneas (**Fig. 6b** and **6c**). After 21 days, the total scores of the SFs and non-treated groups were markedly higher than the qCSKs group ($p=0.001$ and $p=0.001$, respectively) (**Fig. 6e** and **Table S17**). Quantitatively, at PID21, the haze density after qCSK injection was significantly reduced compared to the SF injection ($p=0.006$) and non-treated corneas ($p=0.001$) (**Fig. 6f** and **Table S18**). In the latter 2 groups, although the haze density decreased, the haze areas appeared to enlarge over time (**Fig. 6g** and **Table S19**). The haze areas of both groups were greater than that after the qCSK injection at PID21 ($p=0.001$ vs. SFs and $p=0.068$ vs. non-treated). Unlike in the acute haze model, the CCT was close to the pre-operative value after 21 days in the qCSKs and non-treated groups (**Fig. 6h** and **Table S20**). In contrast, the edema did not subside after SF injection, which resulted in $13.92\pm 7.97\%$ thicker corneas than in the preoperative state.

Similar to the acute haze model, the therapeutic efficacy of qCSKs was significantly better than the SFs and non-treatment in the established haze model. However, the treatment potency was marginally lower in chronic opacity. For example, the median haze and NV scores were 1 (0.5 in the acute opacity model) and 2 (0.5 in the acute opacity model), respectively. Other parameters, such as haze density ($9.93\pm 0.52 \times 10^6$ a.u. vs. $8.19\pm 0.28 \times 10^6$ a.u. in the acute opacity model) and haze area ($28.37\pm 7.35\%$ vs. $7.42\pm 1.09\%$ in the acute opacity model) were also higher in the corneas with established haze.”

The following discussion on the outcome of the chronic opacity model was added to the Discussion (page 37, lines 751-775): “We further investigated qCSK cell injection therapy in a chronic haze model to determine if the therapeutic temporal window could be extended beyond that of acute haze. The pool of patients affected by chronic haze worldwide is large, many of whom reside in medically under-served regions with trachoma as a leading etiology for the disease.² Cell therapy for the treatment of chronic haze would be an efficient use of an already scarce (donor corneas). Our preliminary end-point result of partial scar resolution in a chronic haze model has been notably encouraging and further research is currently ongoing. It is important to note that the current research does not

encompass the entire scope of challenges in chronic corneal haze and scarring. Chronic haze, as compared to acute haze, likely possesses a different microstructural makeup, cellular microenvironment, and immunological landscape as compared to acute scars. Empirically, in our chronic haze model, increased vascularization into the cornea, accompanied by dense stromal haze has been observed. Investigation of the exact pathophysiology of the evolution of a chronic scar remains outside the purview of this study, but we acknowledge that further research in these areas is essential for developing comprehensive therapies for chronic corneal haze and scarring. Two different approaches to tackling chronic haze will be explored in the future. First, because injected qCSKs have a half-life of 3 weeks, an additional injection of qCSKs after 3 weeks may be required to resolve the haze further. Because of the prominent involvement of the immunological component, it is also possible that the remaining scarred tissue is not repairable by the qCSKs. The second strategy, hence, involves the removal of scar tissue, which will then be replaced by a biomaterial, such as collagen or hyaluronic acid,^{42,43} encapsulated with qCSKs. The biomaterial would act as a temporary scaffold, permitting time for the qCSKs to secrete the ECM proper and organize the collagen fibers.”

2. How reproducible is the “irregular” PTK model in their hands in terms of the degree of opacity?

Author Response: Thank you for pointing this out. We appreciate your observation regarding the reproducibility of our model. Within this study alone, the model's reproducibility can be evidenced by the consistency of the haze score at post-injection day 0 across different experimental groups performed on two different days. In the first experiment, as illustrated in Fig. S2, 61.1% of the rats scored 2, and 22.2% scored 3. In the second experiment, corresponding to Fig. 2, 62.5% of the rats scored 2, and 29.2% scored 3. The similarity in haze scores between these experiments, conducted on different days, demonstrates the reproducibility of our model. Additionally, statistical analysis indicates no significant difference in score distribution between the two experiments ($p=0.439$), further confirming the consistency of our results. Further to that, the consistency of the opacity induced by IrrPTK can also be seen in our previous paper, published in *PLoS One* 2013;8:e81544.

3. Regarding lines 843-844, enhancing image colors, particularly red, might give a misleading representation. A better approach would be to use high-quality images without enhancements, or if enhancement is necessary, to provide a detailed method of enhancement.

Author Response: Thank you for pointing this out. We acknowledge and appreciate the reviewer's concern regarding the potential for misleading representation through color enhancement in images. To clarify, the grading was conducted using high-resolution, high-quality images. The purpose of the enhancement was solely to augment the visibility of blood vessels for readers, addressing the challenge of discerning these details in non-enhanced slit-lamp photographs, particularly when viewed in printed format. Nevertheless, we have included a description of the enhancement method used, detailing the specific adjustments made using the ImageJ software in the Methods (page 46, lines 987-993). This addition is also copied below:

“Separately, the red color of the images was enhanced to aid the visualization of blood vessels in the corneas by setting the minimum and maximal red saturation channel to 90 and 140, respectively during color balancing using ImageJ software (National Institutes of Health, Bethesda, MD). The purpose of the enhancement was solely to augment the visibility of blood vessels for readers, addressing the challenge of discerning these details in non-enhanced slit-lamp photographs, particularly when viewed in printed format.”

4. Can the authors comment on feasibility and cost-effectiveness relative to other cell therapy techniques? Can keratocyte culture be scaled up for clinical application?

Author Response: Thank you. In response to the reviewer's query about the feasibility, cost-effectiveness, and scalability of our keratocyte culture method relative to other cell therapy techniques, and its potential for clinical application. We have provided some further clarification of the response in the Discussion (page 39, lines 808-825), which is copied below:

“Our qCSK culture technique offers scalability, feasibility, and potential cost-effectiveness for clinical application. Our current protocol enables propagation of qCSKs, yielding a minimum of 2 million cells in a T75 flask at passage 4 per donor cornea, enough for over 12 treatments at a projected per injection cell count of 150,000 cells. This efficiency, coupled with the use of otherwise non-transplant grade tissue, enhances resource utilization and cost-effectiveness. The overall cost-effectiveness is a multifactorial consideration, encompassing the expenses related to the procurement of tissue, culture media, consumables, labor, regulation, and distribution. Given the nascent stage of our research within the rapidly evolving sector of Advanced Therapy Medicinal Products and cell therapy, a conclusive assessment of economic viability in clinical applications remains an estimate. However, given a similar capital outlay and raw material requirement as corneal endothelium cell therapy—a separate endeavor by our lab—and not accounting for

synergism, similar cost estimations of \$880 for cell therapy vs. \$3710 for transplantation are likely.⁴⁹ Ongoing research is focused on refining cost evaluations, with preliminary comparisons suggesting significant cost advantages over conventional transplantation methods. Additionally, the study of utilizing post-cryopreserved qCSKs for cell therapy is ongoing alongside Good Manufacturing Practice conversion of materials and reagents.”

5. Were the cells injected post-preparation without any freezing? Is this a practical approach for clinical application?

Author Response: We appreciate your attention to this critical aspect of our study. In the current research, we indeed utilized cells immediately post-preparation without freezing. This decision was primarily based on our initial focus on evaluating the immediate biological and therapeutic characteristics without additional confoundment by various factors that can alter the cell state and viability, e.g., choice of GMP grade cryopreservation media, freeze and thaw time, minimum freezing cell count, short term and long term cryopreservation viability. Regarding the practicality of clinical applications, we note the significance of cryopreservation for broader clinical utility. We would also like to note that in our intended use case, there is a step of expansion and serum starvation post cryopreservation P3 which mimics immediate post-preparation conditions and reduces the risk of injecting post-thaw non-viable qCSKs. As we are currently optimizing for GMP-grade cryopreservation protocols, we will certainly look into the viability of using cryopreserved qCSKs and the effect of cryopreservation on the qCSK phenotypes and report it in the next publication. We briefly mentioned this intention in the Discussion (page 39, lines 823-825). Preliminary observations from our early experiments suggest that cells thawed from cryopreserved P3 culture had little to no deviation in morphology and growth as compared to non-cryopreserved cells. This requires further characterization following full GMP conversion of culture and cryopreservation reagents as well as optimization of freeze-time viability, thaw viability, and other variables that may impact cell manufacturing.

6. The injection of 4×10^4 cells in 2 μ l of 1x PBS leads to large errors, hence minor loss during injection, evident from the provided videos, could impact the results. Did they consider using a 5ul volume instead?

Author Response: Thank you for pointing this out. We have optimized the volume of fluid that a rat cornea can expand to in a pilot study. Two μ l was the maximum volume that we could inject via a microliter-graded Hamilton syringe without backflow. While we understand the concern about the potential for errors due to a smaller volume, the decision

to use a 2 µl volume was based on our optimization aimed at maximizing the precision of the cell delivery within the corneal tissue. We believe this approach strikes an optimal balance between the technical limitations of the injection process and the biological constraints of the rat cornea. In a human situation, an injection volume of 150 µl would be viable, given the increase in cornea surface area and thickness compared to a rat, hence, eliminating this issue.

7. Was there an attempt to render the SFs quiescent? In other words, what if SFs are also kept in serum free media for 10 days before administration. An exploration of this could deepen the study's scope.

Author Response: We think this is an excellent suggestion and we have performed additional experiments to address this query. The serum-starved stromal fibroblasts, however, did not exhibit typical keratocyte markers and morphology. The cell culture results have been added to Fig. S1 and described in the Results (pages 10-11, lines 226-241) as follows: “Given that the propagation and cultivation of aCSKs in 0.5% serum is both time-consuming and laborious for cell manufacturing, we aimed to expedite the process by culturing donor-derived stromal cells in 10% serum, followed by serum starvation for 14 days in serum-free complete media, in an effort to generate qCSKs. Our findings revealed that these serum-starved SFs had similar levels of *ALDH1A1* ($p=0.054$), *KERA* ($p=0.129$), and *LUM* ($p=0.484$) to the SFs (**Fig. S1a**). Notably, *ALDH3A1* expression in the serum-starved SFs was markedly higher than that in SFs ($p=1.00 \times 10^{-6}$) but the *CHST6* was substantially lower in the serum-starved SFs ($p=0.019$) and *B3GNT7* was undetectable in both cell types. In terms of morphology, the serum-starved SFs appeared to be in an intermediary state between SFs and qCSKs, characterized by long cell processes but with larger cell bodies, and bipolar and parallel orientation (**Fig. S1b**). Consistent with the gene expression, the protein expression of lumican and keratocan was absent but the *ALDH3A1* exhibited increased expression in the serum-starved cells as compared to SFs (**Fig. S1c**). Due to the inability to recapitulate qCSK morphology and phenotypes, serum-starved SFs were deemed unsuitable as an alternative to qCSKs.”

8. Clarification is needed about the injection site concerning the PTK injury site. Was the injured area targeted, or was it the surrounding region?

Author Response: Thank you for pointing this out. Our approach was to administer the injections paracentrally (edge of the injured site). We mentioned this in the Methods (page 44, lines 939-942 and page 45, lines 969-972). This decision was based on the limited

surface area of the rat cornea. Despite the paracentral administration, the fluid dynamics within the corneal structure enable the cell-containing solution to disperse centrally which can be visualized by AS-OCT (Fig. S2). As a result, the injected cells effectively reached and covered the central IrrPTK injury site. This technique ensured comprehensive cell delivery to the target area while minimizing potential disturbance to the uninjured tissue.

9. The percentage of "unsuccessful" deliveries, as depicted in Fig 2, should be elucidated. Out of the 18 rats, how many had unsuccessful deliveries?

Author Response: We acknowledge the reviewer's request for clarification regarding the percentage of unsuccessful deliveries. We should emphasize that the "poor" delivery was not intentionally created. The 18 rats were from one of our early qCSK injection attempts. Because the results were highly variable, we went over the procedure videos and realized the qCSK therapy outcomes were highly dependent on the cell delivery quality. In our study, each of the 18 rats was assigned to either the "poor" or "good" delivery category, with 9 rats in each. This distribution has now been explicitly stated (page 12, lines 257 and 261) in the revised manuscript to enhance the transparency and understanding of the delivery outcomes.

10. Including unsuccessful deliveries in statistical evaluations may not serve the paper's intent. Given their inherent variability from multiple tries for injection to perforation, I recommend their exclusion from the study.

Author Response: We appreciate the reviewer's suggestion and have accordingly adjusted our approach. We should emphasize that the "poor" delivery was not intentionally created. The 18 rats were from one of our early qCSK injection attempts. Because the results were highly variable, we went over the procedure videos and realized the qCSK therapy outcomes were highly dependent on the cell delivery quality. Also, while addressing the reviewer's query #13 below, we realized that a typo on the volume expansion after "good" delivery had been made. The "good" quality delivery was supposed to result in 3.8x-4.8x volume expansion, not 1.8x-4.8x as written originally. We have made the necessary changes on page 12, line 259. There was no typo on the volume expansion after "poor" cellular delivery. With this, we surmised that: "The "poor" cellular delivery typically resulted in a lower and bigger range of volume expansion (1.2x-3.3x), due to leakage or backflow of the qCSKs-containing injection medium, which appeared to correlate with the extent of haze clearance efficacy, which was poorer and highly variable in comparison to the "good" delivery outcomes. The experiment highlighted the importance of cell density in

determining the therapeutic effects of the qCSK injection." The statement has been added to the Discussion (pages 31-32, lines 650-655). Nevertheless, we have moved the figure (previously Fig. 2) to Supplementary Information (now Fig. S2) and the detailed description of the results (previously on pages 12-14, lines 276-312 in the main text) to Supplementary Information (pages 2-3) to maintain the focus of our paper.

11. The authors note, "We established a metric to categorize the efficacy of cell injections as either 'good' or 'poor' procedures based on several observable visual indicators. Our findings indicated that both the quality of cell delivery and the choice of cell type play pivotal roles in resolving corneal haze and restoring visual function, with qCSKs in conjunction with 'good' quality injections yielding the most favorable outcomes." (lines 156-160). However, given the attached videos which depict injections with varying cell volumes, including those of poor quality might not be conducive to the central premise of the study especially concerning the limited number of 18 rats. In general, I would recommend a large number of animals.

Author Response: We acknowledge the reviewer's observations on the categorization of injection quality in our study. While addressing the reviewer's query #13 below, we realized that a typo on the volume expansion after "good" delivery had been made. The "good" quality delivery was supposed to result in 3.8x-4.8x volume expansion, not 1.8x-4.8x as written originally. We have made the necessary changes on page 12, line 259. There was no typo on the volume expansion after "poor" cellular delivery. With this, we surmised that: "The "poor" cellular delivery typically resulted in a lower and bigger range of volume expansion (1.2x-3.3x), due to leakage or backflow of the qCSKs-containing injection medium, which appeared to correlate with the extent of haze clearance efficacy, which was poorer and highly variable in comparison to the "good" delivery outcomes. The experiment highlighted the importance of cell density in determining the therapeutic effects of the qCSK injection." The statement has been added to the Discussion (pages 31-32, lines 650-655).

On the suggestion to add a larger number of animals, our study's sample size of 18 rats (9 rats in each group) was chosen to achieve at least 80% statistical power. Based on the sample size power calculation, 3 rats in each group were the minimum number of samples to observe a significant difference in haze density, with a calculated mean difference of 1.13 and a standard deviation of 0.25.

12. Considering that corneal regeneration in rats is slower than in mice, a longer follow-up duration than 21 days is advisable.

Author Response: We value your suggestion regarding a longer follow-up period. Our pilot study between 21-56 days indicated no significant changes in healing post-treatment. Moreover, we found that 21 days was adequate to observe the difference in therapeutic effects and collagen fiber changes with qCSK treatment, as opposed to the SF-injected and untreated groups. Hence, this timeframe was emphasized in our report. Nonetheless, we acknowledge the importance of extended studies and will be pursuing so in large animal models, like rabbits and non-human primates.

13. Lines 231-234 detail that a "good" delivery quality is determined by: a single injection entry point and the formation of a cell-containing bleb at the cornea's center, which occupies >17% of the corneal area, typically translating to a 1.8x-4.8x volume expansion. However, in other sections, it's mentioned that a 2µl injection of cells in PBS is the norm. This creates ambiguity regarding whether a successful injection means exactly 2µl or if it's about the resulting expansion (>17% of the corneal area). If these two parameters are equivalent, their correlation should be elucidated. I'd suggest standardizing to intrastromal injections and excluding deviations from this protocol in the study's dataset.

Response: We thank the reviewer for the astute observation of the abnormality in the volume expansion. We went over the raw data and found that the "good" quality delivery was supposed to result in 3.8x-4.8x volume expansion, not 1.8x-4.8x as written originally. We have made the necessary changes on page 12, line 259. There was no typo on the volume expansion after "poor" cellular delivery. With this, we surmised that: "The "poor" cellular delivery typically resulted in a lower and bigger range of volume expansion (1.2x-3.3x), due to leakage or backflow of the qCSKs-containing injection medium, which appeared to correlate with the extent of haze clearance efficacy, which was poorer and highly variable in comparison to the "good" delivery outcomes. The experiment highlighted the importance of cell density in determining the therapeutic effects of the qCSK injection." The statement has been added to the Discussion (pages 31-32, lines 650-655).

14. A viability test is warranted to assess cell survival after passage through a 30-gauge needle.

Author Response: Thank you for your excellent suggestion. We would like to highlight our previous research published in *Invest Ophthalmol Vis Sci* 2018;59:3340–3354, where we

have already investigated this aspect. In that study, we established that qCSKs maintain a high viability rate of over 96% when dispensed through a 30-G needle. However, a significant decrease in cell viability, down to $22\pm 11\%$, was observed when using a finer 33-G needle. We have added the following statement in the Results (page 11, lines 249-252): “We could be certain that it was not due to the cell death induced by shearing force during the dispensing of the cells through a 31G needle. We have previously shown that >96% of the qCSKs were viable when they were ejected through 27, 30, and 31G needles.²²”

15. Lastly, results like the one regarding the adverse outcomes of "poor" delivery (e.g., increased neovascularization) don't substantively advance the study's narrative due to the vast variability in the "poor delivery" category. Like this “Poor” delivery also resulted in increasing neovascularization (NV) severity scores over time (Fig. 2b and 2d and Table S2).

Author Response: Thank you for your valuable feedback. We acknowledge your concern regarding the categorization of “poor delivery” in our study. Recognizing the variability this term encompasses, we have refined this classification, noting that reduced qCSK cell density, often due to leakage or backflow, is a factor in poor therapeutic outcomes such as increased neovascularization. We have moved the figure (previously Fig. 2) to Supplementary Information (now Fig. S2) and the detailed description of the results (previously on pages 12-14, lines 276-312 in the main text) to Supplementary Information (pages 2-3) to maintain the focus of our paper.

Once again, we thank you for the opportunity to address the reviewers' and your comments and revise our manuscript. We look forward to hearing back from you.

Sincerely yours,

Prof. Jodhbir Mehta
Executive Director, Singapore Eye Research Institute
Senior Consultant, Singapore National Eye Centre
Professor, Duke-NUS Medical School

REVIEWERS' COMMENTS

Reviewer #2 (Remarks to the Author):

The authors have added new data that significantly increase the impact of the study. It is acceptable for publication in N Comm.